# Accurate RNA 3D Structure Prediction via Language Model-Augmented AlphaFold 3

## Abstract

Predicting RNA 3D structure from sequence remains challenging due to the structural flexibility of RNA molecules and the scarcity of experimentally resolved structures. We ask how self-supervised RNA language models (LMs), trained on millions of RNA sequences, can best enhance AlphaFold 3 (AF3) for RNA structure prediction. Using an open-source AF3 reproduction, we run controlled experiments that fix data and hyperparameters while varying fusion position and method. We find large performance variations between fusion strategies, and without Multiple Sequence Alignment (MSA), they are generally not effective. When incorporating MSA, the most effective approach is additive fusion applied at the late stage of the conditional network, refining AF3's single representations with RNA LM embeddings. On RecentPDB-RNA (47 newly released targets), our best model achieves an average TM-score of 0.438 and a success rate of 30% (TM-score $\geq$ 0.6), significantly outperforming all baseline models. On 11 CASP16-RNA targets, it matches the best automated system trRosettaRNA. These results show that properly fused RNA LM features substantially advance RNA 3D structure prediction. We will release the data, code, and model weights to support open science, reproducibility, and the development of automated RNA structure prediction models.

## 1 Introduction

Accurately predicting 3D structures of RNA molecules from primary sequences is a remaining grand challenge in biology. It is an important step towards understanding the diverse functions of RNA molecules. It also holds great promise for developing RNA-related therapeutics, such as mRNA vaccines, anti-sense oligonucleotide (ASO) and aptamers Zhu et al. (2022); Androsavich (2024). In recent years, AlphaFold has transformed computational protein structure modeling, achieving predictions with near-experimental accuracy Jumper et al. (2021); Abramson et al. (2024). By contrast, RNA structure prediction remains far more challenging. RNA molecules, composed of only four nucleotides, are inherently more dynamic and flexible than proteins, making experimental determination substantially harder. As of July 8, 2025, the Protein Data Bank (PDB) contains only a few thousand RNA structures, the number of which is less than 5% of the number of deposited protein structures Berman et al. (2000). Due to the scarcity of experimentally determined RNA structures, RNA 3D structure prediction becomes a small data high-dimensional machine learning problem. As measured in the Critical Assessment of protein Structure Prediction (CASP) 16 blind competition, all the top-performing groups for RNA structure prediction are human expert predictors Kretsch et al. (2025). The reliance on manual expertise in modeling each RNA structure, however, significantly limits the prediction speed and application scope, particularly in the scenario of drug candidate screening.

In this work, we seek an automated model for accurate RNA 3D structure prediction. For a given RNA nucleotide sequence, an automated model can directly output the 3D coordinate prediction for each atom in the RNA molecule without being further processed by human experts. Computational methods have been developed for more than two decades. Early approaches to RNA structure prediction primarily rely on physics-inspired energy functions to simulate molecular folding. Template-based methods, which resemble retrieval strategies, are later introduced to leverage homologous RNA structural information. More recently, with the rise of deep learning and particularly following the success of AlphaFold, deep learning–driven approaches have attracted increasing attention and are

rapidly reshaping the field. Methods such as trRosettaRNA Wang et al. (2023), RhoFold+ Shen et al. (2024) and NuFold Kagaya et al. (2025) are very much inspired by AlphaFold 2, while recently AlphaFold 3 extends its predictions to different molecules including RNA. Systematic benchmarking shows that AF3 is a competitive method that outperforms most of the existing solutions for RNA 3D structure prediction (Bernard et al., 2025).

In parallel with advances in RNA structure modeling, progress in RNA sequence modeling has driven the development of increasingly powerful RNA language models (LMs). Through self-supervised learning on tens of millions of RNA sequences, RNA LMs capture evolutionary and structural information, achieving impressive performance across diverse RNA function and structure prediction tasks Shen et al. (2024); Penić et al. (2025); Zou et al. (2024). A natural question is: can representations learned from massive RNA sequences by RNA LMs be leveraged to enhance AF3's performance on RNA 3D structure prediction? The motivation is that, although AF3 is jointly trained on protein, RNA, and DNA structural data, proteins dominate the training set. As a result, RNA-specific representations may be underdeveloped and could benefit from the richer features provided by RNA LMs.

To answer this question, we are facing a multimodal fusion problem, integrating information from multiple modalities with the goal of predicting an RNA structure. The technical challenges for multimodal fusion are: 1) representations from RNA LM and AF3 are not in the same feature space; and 2) it is difficult to build models that exploit supplementary and not only complementary information Baltrušaitis et al. (2018). The high complexity of AF3's architecture further complicates the problem. There are five positions in AF3, lying in upstream and downstream of the network, that can be good candidates for feature fusion. For each position, there are several fusion methods, such as add fusion and attention-based fusion, that can be used. It is unclear how to best incorporate RNA LM's representation into AF3.

To investigate this, we design a series of controlled experiments on fusing RNA LM's representation into AF3 [1], keeping the training data and hyperparameters fixed while varying only the fusion positions and methods. We evaluate these models on RecentPDB-RNA, a curated test set comprising 47 RNA targets from the PDB, each released after the training data temporal cutoff and filtered to ensure a maximum sequence similarity of 0.8 to the training set. We find large performance variations between fusion strategies, and without MSA, they are generally not effective. When incorporating MSA, the most effective approach is additive fusion applied at the late stage of the conditional network, refining AF3's single representations with RNA LM embeddings. On 47 RecentPDB-RNA targets, our best fusion model achieves a state-of-the-art in RNA 3D structure prediction, with an average TM-score of 0.438 and a success rate of 30%. On 11 CASP16-RNA targets released in 2025, it surpasses most of the baselines, reaching the performance of the best automated method in the CASP16 competition. These results demonstrate that the representations learned from RNA LMs are informative for RNA 3D structure prediction when incorporated using the right strategy.

## 2 RELATED WORK

### 2.1 RNA 3D STRUCTURE PREDICTION METHODS

Computational modeling of RNA 3D structures, which seeks to predict the atomic positions of nucleotides, has been studied for over two decades. Existing approaches can be broadly categorized into three groups: *ab initio*, template-based, and deep learning–based methods Bernard et al. (2024).

*Ab initio* methods simulate the underlying physics of RNA folding by optimizing energy functions through sampling Boniecki et al. (2016); Zhang et al. (2021); Li & Chen (2023). While physically motivated, these approaches face two key limitations: (1) the simulations are computationally expensive, particularly for large RNAs, and (2) inaccuracies in the energy function can bias sampling and yield incorrect predictions.

Template-based methods leverage the principle that evolutionarily related molecules often adopt similar structures. They construct models using global and local structural information from experimentally solved homologous RNAs Cao & Chen (2011); Popenda et al. (2012); Li et al. (2022).

---

[1]Due to the license of AF3, we use a fully open-sourced reproduction called Protenix Team et al. (2025) in our experiments.

When suitable templates are available, these methods can be highly accurate. However, they are constrained by template availability, which is often lacking for designed or novel RNA sequences.

Deep learning–based methods have recently emerged as powerful alternatives. These approaches train neural networks to predict RNA 3D structures from sequences and/or multiple sequence alignments (MSAs). Based on scope, they can be divided into RNA-specific and general methods. RNA-specific models include DeepFoldRNA Pearce et al. (2022), trRosettaRNA Wang et al. (2023), DRfold Li et al. (2023), RhoFold+ Shen et al. (2024), NuFold Kagaya et al. (2025), and DRfold2 Li et al. (2025). Among these, the first three employ hybrid strategies, combining deep learning for feature learning with energy minimization for final refinement, while the latter three adopt end-to-end architectures inspired by AlphaFold 2 Jumper et al. (2021). General-purpose approaches include RoseTTAFoldNA Baek et al. (2024), RoseTTAFold All-Atom Krishna et al. (2024), AF3 Abramson et al. (2024), and its reproductions such as Protenix Team et al. (2025), Boltz-1 Wohlwend et al. (2024), and Chai-1 team et al. (2024). Among them, AF3 currently delivers SOTA performance across diverse macromolecular assemblies, but its usage is strictly limited by its license.

## 2.2 Incorporating Language Models Into Structure Prediction

The integration of pretrained LMs into structure prediction has gained significant attention in recent years due to the huge success of large language models. In the protein domain, ESMFold Lin et al. (2023) and HelixFold-single Fang et al. (2023) demonstrate that large-scale pretrained protein LMs can substitute for MSAs, achieving performance close to AlphaFold 2 while providing substantially faster inference.

In RNA, recent studies have begun to explore similar directions, not to eliminate MSAs but to improve structural accuracy. RhoFold+ Shen et al. (2024) and DRfold2 Li et al. (2025) both incorporate pretrained RNA LMs and report strong improvements in RNA 3D structure prediction. Notably, RhoFold+ retains both the RNA LM and MSA modules, representing a hybrid approach rather than a full replacement.

In this work, we extend AF3's RNA structure prediction capability by incorporating RNA LM representations, emphasizing the enhancement of RNA representation quality in AF3 or AF3-like architectures through effective multimodal fusion.

## 3 Preliminary

AF3 is a diffusion-based generative model that, conditioned on primary sequences and optional inputs such as multiple sequence alignments (MSAs), predicts all-atom 3D coordinates of biomolecules. For example, for an RNA primary sequence, given noisy coordinates of all the atoms in the RNA molecule, it iteratively denoises them into physically plausible conformations by conditioning on the sequence. Most computation resides in the conditioning network, which takes the primary sequence as input and produces rich single- and pair-wise features that guide the diffusion sampler. As shown in Figure 1, the overall architecture of AF3 contains:

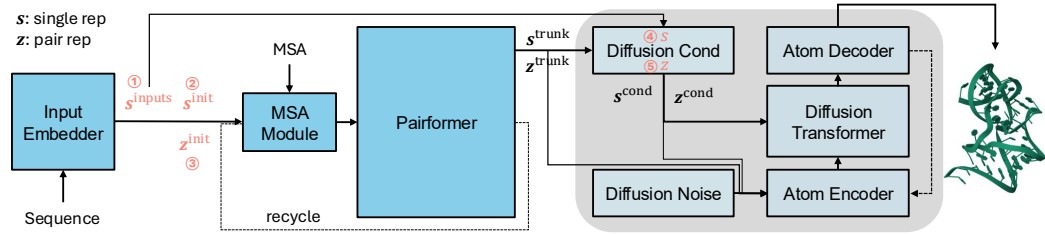

Figure 1: Overview of AF3-style model architecture and the information flow from sequence to single representation and pair representation and to final atomic structure. The grey block denotes the Diffusion Module. The salmon color indicates candidate positions for feature fusion.

**Input Embedder**: A small Transformer embeds the tokens in the primary sequence of length $N_{\text{token}}$ into *single* representations $\mathbf{s}^{\text{inputs}} \in \mathbb{R}^{(N_{\text{token}}, c_{s\_\text{inputs}})}$, and produces an initial *single* representation $\mathbf{s}^{\text{init}} \in \mathbb{R}^{(N_{\text{token}}, c_s)}$ by a linear projection and a *pair* representation $\mathbf{z}^{\text{init}} \in \mathbb{R}^{(N_{\text{token}}, N_{\text{token}}, c_z)}$ by the outer sum of the single representation as inputs for the following Pairformer blocks.

**Pairformer**: A large trunk jointly updates the single representation $\mathbf{s}$ and pair representation $\mathbf{z}$ using attention with geometric interactions. The trunk stacks 48 blocks that exchange information between $\mathbf{s}$ and $\mathbf{z}$ and injects structural priors (e.g. triangular inequality), producing conditioning features $\mathbf{s}^{\text{trunk}} \in \mathbb{R}^{(N_{\text{token}}, c_s)}$ and $\mathbf{z}^{\text{trunk}} \in \mathbb{R}^{(N_{\text{token}}, N_{\text{token}}, c_z)}$ tailored for coordinate generation.

**Diffusion Module**: A conditional, non-equivariant generative model works on the point cloud of atoms. Conditioned on the trunk outputs $\mathbf{s}^{\text{trunk}}$ and $\mathbf{z}^{\text{trunk}}$ and the Input Embedder output $\mathbf{s}^{\text{inputs}}$, a denoising diffusion module iteratively refines noisy atomic coordinates to a final structure (distribution). It is a two-level architecture, operating first on atom-level, then on token-level, and then on atom-level to produce atom-level coordinate predictions.

## 4 METHODOLOGY

We extend AF3 to investigate whether self-supervised representations learned from millions of RNA sequences can enhance RNA 3D structure prediction. To this end, we incorporate RNA LM embeddings into AF3, systematically exploring different fusion positions and methods. The basic idea is to refine AF3's single or pair representations using RNA LM embeddings.

### 4.1 FEATURE EXTRACTION FROM RNA LM

Given an RNA sequence of $N_{\text{token}}$ nucleotides, let $\mathbf{s}^{\text{rnalm}} \in \mathbb{R}^{(N_{\text{token}}, c_{\text{rnalm}})}$ denote the final-layer hidden states from the RNA LM, where $c_{\text{rnalm}}$ is the embedding dimension. We lift these single-token embeddings to pair space by forming $\mathbf{z}^{\text{rnalm}} \in \mathbb{R}^{(N_{\text{token}}, N_{\text{token}}, c_z)}$ by projecting $\mathbf{s}^{\text{rnalm}}$ twice to $c_z$ channels and computing an outer sum between the two projected matrices, where $c_z$ is the pair representation embedding dimension in AF3. In specific,

$$\mathbf{z}^{\text{rnalm}}_{ij} = \mathbf{s}^{\text{rnalm}}_i W_1 + \mathbf{s}^{\text{rnalm}}_j W_2$$

where $W_1, W_2 \in \mathbb{R}^{(c_{\text{rnalm}}, c_z)}$ are trainable parameters, and $i, j$ denote positions in the sequence. The outer-sum construction yields a symmetric pair representation, i.e., $\mathbf{z}^{\text{rnalm}}_{ij} = \mathbf{z}^{\text{rnalm}}_{ji}$, when the two projections are tied ($W_1 = W_2$).

### 4.2 MULTIMODAL FUSION STRATEGIES

**Fusion positions**  By diving into the architecture of AF3, we locate five candidate positions for feature fusion:

1. The input single representation $\underline{\mathbf{s}}^{\text{inputs}}$;
2. The initial single representation $\underline{\mathbf{s}}^{\text{init}}$;
3. The initial pair representation $\underline{\mathbf{z}}^{\text{init}}$;
4. The single conditioning representation $\underline{\mathbf{s}}$ in diffusion module;
5. The pair conditioning representation $\underline{\mathbf{z}}$ in diffusion module.

As shown in Figure 1, among the five candidate fusion positions, the first three lie upstream of the Pairformer; features fused at these locations are subsequently processed by the Pairformer. The remaining two positions are downstream of the Pairformer; features injected there bypass it and are used directly to condition the Diffusion Module. To avoid redundancy, we fuse at a single position per model variant rather than at multiple positions simultaneously.

**Fusion methods**  For fusion methods, we adopt commonly used methods as candidates:

1. Add Fusion: add RNA FM's embedding (or its outer concatenation) to the targeted representation;

2. Concat Fusion: concatenate RNA FM's embedding with the targeted representation along the feature dimension;

3. Cross-attention Fusion: treat the targeted representation as a query and RNA FM's embedding as key and value, and use the multi-head cross attention mechanism Vaswani et al. (2017) to extract information from the RNA FM's embedding to the targeted representation.

Note that Concat Fusion will change the targeted representation's dimension, while Add Fusion and Cross-attention Fusion do not change the targeted representation's dimension.

**Fusion strategies**   For single representation $\mathbf{s}_{af} \in \mathbb{R}^{(N_{\text{token}}, c)}$, the updated single representation after feature fusion is

$$\mathbf{s}_{af} = \begin{cases} \sigma\left(\mathbf{s}^{\text{rnalm}} W_2\right) \odot \mathbf{s}_{af} + \mathbf{s}^{\text{rnalm}} W_1, & \text{if Add Fusion,} \\ [\mathbf{s}_{af}; \mathbf{s}^{\text{rnalm}}], & \text{if Concat Fusion,} \\ \text{CrossAttention}(q = \mathbf{s}_{af},\ kv = \mathbf{s}^{\text{rnalm}}), & \text{if Cross-attention Fusion.} \end{cases} \quad (1)$$

where $W_1, W_2 \in \mathbb{R}^{(c_{\text{rnalm}}, c)}$, $\sigma(.)$ is a sigmoid function. When the fusion happens in the Diffusion Conditioning Module, we do not use the gate function for Add Fusion, so it becomes: $\mathbf{s}_{af} = \mathbf{s}_{af} + \mathbf{s}^{\text{rnalm}} W_1$.

For pair representation $\mathbf{z}_{af} \in \mathbb{R}^{(N_{\text{token}}, N_{\text{token}}, c_z)}$, the updated pair representation after feature fusion is

$$\mathbf{z}_{af} = \begin{cases} \mathbf{z}_{af} + \mathbf{z}^{\text{rnalm}}, & \text{if Add Fusion,} \\ [\mathbf{z}_{af}; \mathbf{z}^{\text{rnalm}}], & \text{if Concat Fusion.} \end{cases} \quad (2)$$

For the detailed algorithms, please refer to the Appendix Section A.3.

## 5 EXPERIMENTS

### 5.1 TRAINING DATA

For training data, we use RNA3DB, a curated collection of structured RNAs derived from Protein Data Bank Szikszai et al. (2024). The following chains were excluded: 1) shorter than 32 residues; 2) with structural resolution higher than 9Å; 3) a single nucleotide makes up more than 80% of residues; and 4) more than 30% of the residues are "unknown". The remaining RNA chains were clustered at 99% sequence identity. RNA3DB preserves all the chains in the cluster since they are associated with different experimentally determined structures. While the chain is the same, it is possible that the presence of different interacting partners in the actual crystal structures may result in different structural conformations. RNA3DB preserves the full extent of the structural diversity present in PDB. We use the 2024-12-04 release of RNA3DB, comprising 12,892 samples spanning 2,687 unique RNA chains with approximately 5 structures per chain. The average sequence length for the unique sequences is 742. 30% of the data have a sequence length over 384. For multiple sequence alignments (MSAs), we retrieve them from `MSA_v2` data from Stanford RNA 3D Folding He et al. (2025a) searched by rMSA Zhang et al. (2023), which covers 39% of the training sequences.

### 5.2 EVALUATION DATA

**RecentPDB-RNA evaluation set**   We collected RNA structures from the PDB released between December 4, 2024 and April 28, 2025, selecting entries with a resolution better than 4Å and RNA sequence lengths between 30 and 1,000 nucleotides. Each complex contains no more than 20 RNA chains, yielding 67 unique sequences. After filtering out samples with sequence similarity ≥80% to the training set, 54 sequences remained. We then cross-checked these PDB entries against the latest RNA3DB release (2025-10-01-incremental-release) and successfully retrieved 150 corresponding structures for 47 of the sequences, with each sequence corresponding to an average of 3 structures. The average sequence length is 242, with a minimum length of 36 and a maximum length of 814. We search MSA for these targets using rMSA Zhang et al. (2023), an automated pipeline that searches

and aligns homologs from RNAcentral, Rfam, and nt databases (see Appendix Table 2 for database versions) for a target RNA. The distributions of sequence length, sequence similarity between the test and training sets, and the number of effective sequences (Neff) in the MSAs are provided in Appendix Table 1.

**CASP16-RNA evaluation set**  We collected the CASP 16 RNA targets with experimental structures released in PDB in 2025, containing 11 targets in total. The target ids are: R1205, R1209, R1211, R1242, R1263v1, R1264v1, R1286, R1251, R1283v1, R1296, R1285. The MSA retrieval is the same as described in the above section. The average sequence length is 288, with a minimum length of 59 and a maximum length of 833. For distributions of the sequence length, sequence similarity to the training set, and Neff of MSA, please refer to Appendix Table 1.

## 5.3 EXPERIMENTAL SETTING

**Training setting**  Due to the license of AlphaFold 3, we cannot use it and instead, we use a successful and fully open-sourced reproduction called Protenix [2] as the backbone for our experiment Team et al. (2025). In specific, we use the Protenix released checkpoint `model_v0.2.0.pt` for all of our experiments. For the RNA foundation model, we use AIDO.RNA, a strong transformer-based encoder-only language model pretrained on 42 million non-coding RNA sequences from RNAcentral Zou et al. (2024). In specific, we use `AIDO.RNA-650M` through `AIDO.ModelGenerator` Ellington et al. (2025). We train all the models on the RNA3DB dataset with AIDO.RNA frozen whenever it is used. For models with different fusion strategies, we use the same training setting for fair comparisons. We use a two-stage training, with Stage 1 to warm up the newly initialized weights (adapters) while keeping Protenix frozen and Stage 2 to jointly train Protenix and the adapters. We apply an exponential moving average (EMA) to the model weights with a decay rate of 0.999. We freeze the confidence head and increase the diffusion trunk size to accelerate the training process. We also train two baseline models (with and without MSAs) without any feature fusion using the same setting to understand how the training data contributes to the performance. The detailed training hyperparameters are listed in Appendix Table 4. The global batch size is set to 16, with a micro batch size of 1 and gradient accumulation steps of 4. For each experiment, training was performed on 4 NVIDIA A100-80GB GPUs with distributed data parallel. The training time for each experiment is about 2.5 days.

**Inference setting**  We use the default inference setting in Protenix, with the last EMA checkpoint for each experiment. Note that the Protenix checkpoint was not trained with RNA MSAs. For models trained without RNA MSAs, we do not use MSAs in inference. For models trained with RNA MSAs, MSAs are utilized during inference. Detailed inference hyperparameters are listed in the Appendix Table 5.

**Evaluation metrics**  Following common practice, we use TM-score (Template Modeling Score) as our major evaluation metric, which is used to assess the structural similarity between the predicted structure and the ground truth structure. It ranges from 0.0 to 1.0, with a higher value indicating a better prediction. A prediction is considered successful if its TM-score is $\geq 0.6$. For each target in the test set, we generate 5 predictions. The final score is the average of best-of-5 TM-scores of all targets. The TM-score is computed on the C1' atom using the following USalign Zhang et al. (2022) script: `USalign {pred_pdb} {true_pdb} -atom " C1'" -m - -mol RNA -TMscore 1`.

## 6 RESULTS AND ANALYSIS

### 6.1 EFFECT OF RNA LM FEATURE FUSION STRATEGIES WITHIN AF3-LIKE ARCHITECTURE

We conducted controlled experiments to evaluate different fusion strategies for incorporating RNA LM representations into Protenix, varying only the position and method of feature fusion. Since the original Protenix model was not trained with RNA MSAs, we adopted the same setting and first trained 11 models without MSA input, including a baseline without any feature fusion for reference. As shown in Table 1, the original Protenix achieves a TM-score of 0.325 with 3 successful predictions

---

[2] https://github.com/bytedance/Protenix

Table 1: **RNA 3D structure prediction performance of RNA LM fusion strategies in an AF3-like architecture (Protenix) on RecentPDB-RNA.** Bold indicates the best result and underline indicates the second best results.

| | Fusion position | Fusion method | Use MSA | TM-score ↑ | #Success ↑ |
|---|---|---|---|---|---|
| Original Protenix Team et al. (2025) | | | ✘ | 0.325 | 3 |
| Finetuned Protenix | none | none | ✘ | 0.415 | 11 |
| | | none | ✔ | 0.399 | 11 |
| RLM-aug Protenix | inputs $\mathbf{s}^{\text{inputs}}$ | add | ✘ | 0.382 | 11 |
| | | cross attention | ✘ | 0.376 | 8 |
| | init single rep $\mathbf{s}^{\text{init}}$ | add | ✘ | 0.419 | 11 |
| | | add | ✔ | 0.409 | 10 |
| | | concat | ✘ | 0.406 | 10 |
| | | cross attention | ✘ | 0.412 | 11 |
| | | cross attention | ✔ | 0.409 | 11 |
| | init pair rep $\mathbf{z}^{\text{init}}$ | add | ✘ | 0.370 | 9 |
| | single conditioning $\mathbf{s}$ | add | ✘ | 0.397 | 12 |
| | | add | ✔ | **0.438** | **14** |
| | | concat | ✘ | 0.360 | 10 |
| | pair conditioning $\mathbf{z}$ | add | ✘ | 0.402 | 10 |
| | | concat | ✘ | 0.393 | 10 |

out of 47 targets on RecentPDB-RNA. Finetuning Protenix without MSA on RNA3DB substantially improves performance, increasing the TM-score to 0.415 and the number of successful predictions to 11. In comparison, RLM-aug Protenix models show diverse performances, highlighting that different fusion strategies have varying effects on RNA 3D structure prediction.

Given the proven effectiveness of MSAs in protein structure prediction and the fact that AF3 was trained with RNA MSAs, we further trained four models with RNA MSAs: three using the top-performing fusion strategies identified earlier without MSAs—(initial single representation, add), (initial single representation, cross-attention), and (single conditioning, add)—and one baseline model without RNA LM as a reference. As shown in Table 1, the RLM-aug Protenix (single-conditioning, add) trained with MSA achieves a TM-score of 0.438, representing a 10% relative improvement over the Finetuned Protenix with MSA baseline and increasing the success rate from 23% to 30%. One-sided paired t-tests indicate that the RLM-aug Protenix (single-conditioning, add) model trained with MSA performs significantly better than both the Protenix baselines and the other RLM-aug Protenix variants, with statistical significance at $\alpha = 0.05$.

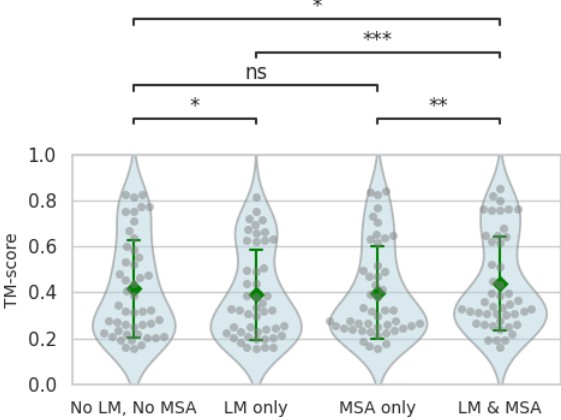

Figure 2: **Effect of RNA LM and MSA in training on RecentPDB-RNA.** Gray dots represent individual TM-scores, and green lines indicate ±1 standard deviations from the mean. All four models are variants of Protenix finetuned on the same dataset, differing only in whether RNA LM or MSA is used during training. For models incorporating the RNA LM, the (single conditioning, add) fusion strategy is used. Two-sided paired $t$-tests were performed between models (ns: $p > 0.05$, *: $0.01 < p \leq 0.05$, **: $0.001 < p \leq 0.01$, ***: $p \leq 0.001$).

**Effect of RNA LM and MSA in training** We observed an intertwined effect between the RNA LM and MSA. To disentangle their contributions, we conducted an ablation study on the finetuned Protenix model with and without LM × with and without MSA. As shown in Figure 2, fine-tuning Protenix with either LM or MSA alone does not yield noticeable improve-

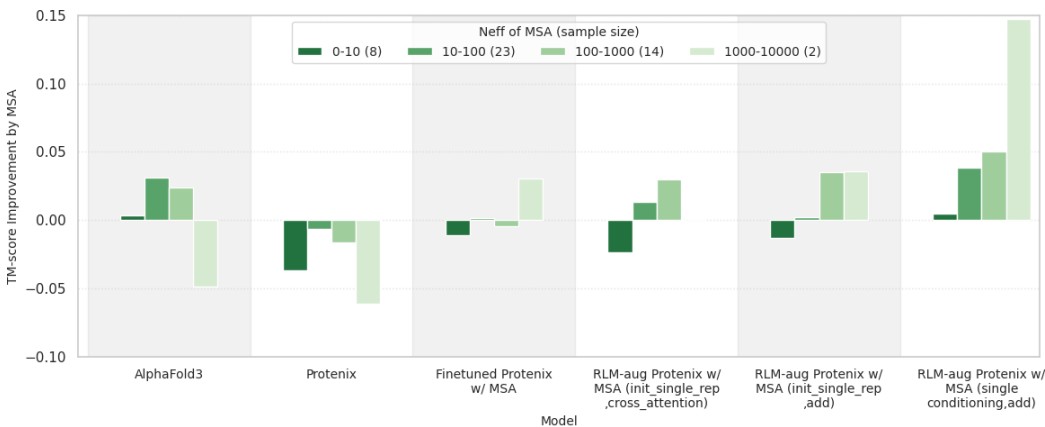

Figure 3: **Effect of MSA in inference on RecentPDB-RNA.** The y-axis denotes the difference of TM-score between using MSAs and not using MSAs during inference for the same model.

ment, whereas combining both leads to a significant increase in TM-score. This demonstrates the complementary relationship between RNA LM and MSA, where their joint use enhances structural prediction performance.

**Effect of RNA LM and MSA in inference**    To assess how RNA MSAs influence model performance during inference, we grouped the RecentPDB-RNA test sequences into four bins based on their effective number of sequences (Neff) in the MSA: 0–10, 10–100, 100–1000, and 1000–10000, containing 8, 23, 14, and 2 samples, respectively. We evaluated AlphaFold 3 (as a reference), Protenix, Finetuned Protenix with MSA, and three RLM-aug Protenix variants with MSA. For each model, we plotted the TM-score improvement obtained by inference with MSA compared to inference without MSA (Figure 3). For the two Protenix models without RNA LM, incorporating MSA generally did not improve performance, especially for low-Neff sequences. In contrast, the three models trained with RNA LM + MSA showed a consistent positive trend: as Neff increased, the performance gain from MSA became more pronounced. Notably, the RLM-aug Protenix (single-conditioning, add) variant exhibited the strongest ability to leverage MSA information, achieving the largest improvements with higher Neff. These results demonstrate that integrating an RNA LM enhances the model's capacity to exploit evolutionary information from MSAs during inference, suggesting that the RNA LM and MSA act synergistically—the LM provides contextual priors that enable the network to make better use of evolutionary features otherwise underutilized in baseline architectures.

## 6.2 BENCHMARKING AGAINST EXISTING RNA STRUCTURE PREDICTION MODELS

In this section, we benchmark our best fusion model, RLM-aug Protenix (single conditioning, add), trained with RNA MSAs against existing automated methods, including AlphaFold 3 and other RNA-specific models. The baseline versions we used are listed in Appendix Table 3.

**Results on RecentPDB-RNA**    As shown in Table 2, among the four deep learning based RNA-specific structure predictors, trRosettaRNA achieves the best performance, with a TM-score of 0.332 and a success rate of 11% on 47 RecentPDB-RNA targets. By contrast, AlphaFold 3 achieves a slightly higher TM-score of 0.358 but a lower success rate of 9%. Protenix attains a TM-score of 0.325 with a success rate of 6%, which is competitive but slightly lags behind AlphaFold 3. Finetuning Protenix on the RNA3DB dataset yields a large performance boost. When incorporating RNA LM (single conditioning, add) and MSA, the model achieves a TM-score of 0.438 and a success rate of 30%, significantly outperforming the other baseline models (one-sided pair t-test with significance level $\alpha = 0.05$), including the two LM-based models, RhoFold+ and DRfold2. For illustration, we visualize the predicted structures of our best model alongside other methods on the test target 8SYK_A in Appendix Figure 1, where the ground truth structure from the PDB is shown in green.

Table 2: **RNA 3D structure prediction results on RecentPDB-RNA and CASP16-RNA test sets.** For the Vfold Pipeline, 14 targets on RecentPBD-RNA and 5 targets on CASP16-RNA failed to return predicted 3D structures. TM-score averages were taken on those with predicted structures. The trRosettaRNA server generates five structural decoys but outputs only the one with minimal free energy. ** denotes RLM-aug Protenix w/ MSA (single conditioning, add) is significantly better than the corresponding baseline model (one-sided pair t-test on RecentPDB-RNA, one-sided Wilcoxon signed-rank test on CASP16-RNA, p-value < 0.05) while "ns" denotes not significant.

| | RecentPDB-RNA (47) | | CASP16-RNA (11) | |
|---|---|---|---|---|
| | TM-score ↑ | Success rate ↑ | TM-score ↑ | Success rate ↑ |
| Vfold (human expert) from CASP16 website | | | 0.486 | 36% |
| Vfold Pipeline* (Li et al., 2022) | 0.279 | 0% | 0.289 | 0% |
| NuFold (Kagaya et al., 2025) | 0.282 ** | 2% | 0.243 ** | 0% |
| RhoFold+ (Shen et al., 2024) | 0.309 ** | 9% | 0.277 ** | 0% |
| DRfold2 (Li et al., 2025) | 0.316 ** | 9% | 0.313 ** | 18% |
| trRosettaRNA* (Wang et al., 2023) | 0.332 ** | 11% | 0.412 (ns) | 27% |
| AlphaFold 3 (Abramson et al., 2024) | 0.358 ** | 9% | 0.371 (ns) | 9% |
| Protenix (Team et al., 2025) | 0.325 ** | 6% | 0.340 ** | 18% |
| Finetuned Protenix w/o MSA | 0.415 ** | 23% | 0.410 (ns) | 27% |
| Finetuned Protenix w/ MSA | 0.399 ** | 23% | 0.421 (ns) | 27% |
| RLM-aug Protenix w/ MSA (init single rep, add) | 0.409 | 21% | **0.451** | 27% |
| RLM-aug Protenix w/ MSA (init single rep, cross-atten.) | 0.409 | 23% | 0.431 | 27% |
| RLM-aug Protenix w/ MSA (single conditioning, add) | **0.438** | **30%** | 0.422 | 27% |

**Results on CASP16-RNA** We further evaluated our models on 11 CASP16-RNA targets, as shown in Table 2. On this benchmark, RLM-aug Protenix w/ MSA (single-conditioning, add) achieves a TM-score of 0.422 and a success rate of 27%, significantly outperforming NuFold, RhoFold+, and DRfold2 (one-sided Wilcoxon signed-rank test, significance level $\alpha = 0.05$). Its performance is also comparable to AlphaFold 3 and trRosettaRNA, the latter ranking first among Server groups in the CASP16 RNA prediction experiment (team name: Yang-Server). The other two RLM-aug Protenix w/ MSA variants achieve TM-scores of 0.451 and 0.431, respectively. However, given the small sample size ($n = 11$), we avoid drawing strong conclusions from this benchmark alone. To assess the current state of the field, we refer to the top-performing Vfold model with human expert input reported on the CASP16 website and observe that a substantial performance gap still exists between automated and expert-guided approaches. Considering the performance of the Vfold Pipeline, it is evident that human expertise remains essential for achieving high-accuracy RNA 3D structure prediction.

## 6.3 ANALYSIS

**Effect of sequence identity** To assess how sequence identity between training and test data influences model performance, we divided the RecentPDB-RNA targets into four categories based on

Table 3: Effect of sequence identity on RecentPDB-RNA. TM-score is reported.

| | Max seq sim to train | | | | All |
|---|---|---|---|---|---|
| | <0.5 (13) | 0.5-0.6 (18) | 0.6-0.7 (11) | 0.7-0.8 (5) | (47) |
| [1] Protenix | 0.225 | 0.333 | 0.458 | 0.266 | 0.325 |
| [2] Finetuned Protenix w/ MSA | 0.242 | 0.356 | 0.592 | 0.539 | 0.399 |
| [3] RLM-aug Protenix w/ MSA (single cond., add) | 0.303 | 0.377 | 0.619 | 0.609 | 0.438 |
| Δ ([2]-[1]) | 0.017 | **0.023** | **0.134** | **0.273** | **0.074** |
| Δ ([3]-[2]) | **0.061** | 0.021 | 0.028 | 0.070 | 0.039 |
| ([2]-[1])/([3]-[1]) (contribution of data+MSA) | 22% | **52%** | **83%** | **80%** | **66%** |
| ([3]-[2])/([3]-[1]) (contribution of RNA LM) | **78%** | **48%** | 17% | 20% | 34% |

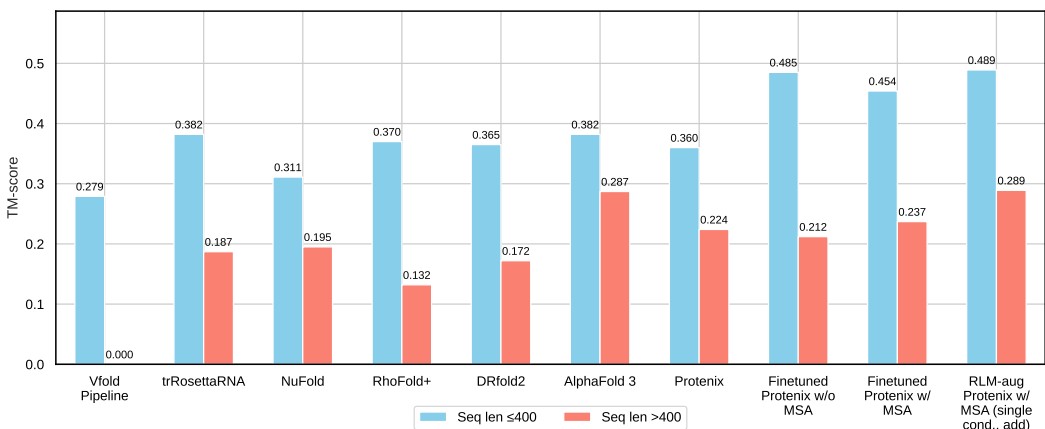

Figure 4: Performance comparison on RecentPDB-RNA by RNA sequence length. The test set is divided into two groups: sequence length $\leq 400$ (35 samples) and $> 400$ (12 samples).

their maximum sequence identity to the training set: $< 0.5$, $0.5$–$0.6$, $0.6$–$0.7$, and $0.7$–$0.8$, containing 13, 18, 11, and 5 samples, respectively. As shown in Table 3, the gap between Finetuned Protenix w/ MSA and the original Protenix reflects the contribution of training data and MSA, while the gap between RLM-aug Protenix w/ MSA (single conditioning, add) and Finetuned Protenix w/ MSA reflects the added benefit of RNA LM representations. Finetuning directly on RNA3DB with MSA substantially improves prediction accuracy for targets with $\geq 0.6$ sequence identity but yields only marginal gains for low-identity ($< 0.6$) cases. In contrast, integrating RNA LM representations enhances performance across all similarity levels, with a large gain observed for low-identity targets. These findings indicate that RNA LM features improve generalization beyond training-like examples.

**Effect of sequence length** To assess the effect of sequence length, we divided the RecentPDB-RNA test set into short ($\leq 400$ nucleotides, 35 samples) and long ($> 400$ nucleotides, 12 samples) targets. As shown in Figure 4, for short sequences, most models perform reasonably well, with RLM-aug Protenix w/ MSA (single-conditioning, add) achieving the highest TM-score of 0.489. For long sequences, performance drops substantially across all models. The Vfold Pipeline fails to generate valid predictions for these targets, while RhoFold+, DRfold2, trRosettaRNA, and NuFold reach TM-scores of 0.132, 0.172, 0.187, and 0.195, respectively, highlighting the significant difficulty of large RNA structure prediction. AlphaFold 3 attains a TM-score of 0.287, demonstrating the strongest robustness to sequence length, likely due to its effective use of RNA MSA and multi-stage training on long sequences. In comparison, RLM-aug Protenix w/ MSA (single-conditioning, add) achieves a TM-score of 0.289, matching AlphaFold 3. When comparing the finetuned Protenix variants, we observe that incorporating either MSA or RNA LM embeddings during training improves performance on long RNA sequences, indicating that evolutionary information plays an important role in modeling large and complex RNA structures.

## 7 CONCLUSIONS AND FUTURE WORK

In this work, we systematically explore strategies for integrating RNA LM representations into an AF3-like architecture to improve RNA 3D structure prediction. Our results show that injecting the LM representation into the single representation of the Diffusion Conditioning Module yields the most effective performance, achieving SOTA or near-SOTA performance on two test sets. Additional analyses further suggest that RNA LMs are particularly beneficial for predicting large RNA structures.

Despite these advances, our approach has several limitations: (1) it is specialized for RNA structure prediction, leaving its applicability to proteins and DNA uncertain; (2) the confidence prediction head was not finetuned, making it an unreliable reference beyond the Protenix version; (3) as a data-driven method, performance strongly depends on the quantity and diversity of training data and the generalization ability to out-of-distribution targets is limited; and (4) the absolute accuracy for large RNA structures remains suboptimal. A natural direction to address the first limitation is to

extend our framework by replacing the RNA LM with multimodal biological language models, such as LucaOne He et al. (2025b), thereby enabling all-atom structure prediction across proteins, RNA, and DNA. We leave this exploration for future work.

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

# A APPENDIX

## A.1 EVALUATION DATA

Table 1: Data distribution of sequence length, max sequence similarity to training set, and number of effective sequences in MSA for RecentPDB-RNA and CASP16-RNA.

|  |  | RecentPDB-RNA | | CASP16-RNA | |
| --- | --- | --- | --- | --- | --- |
|  |  | count | ratio | count | ratio |
| Seq len | ≤400 | 35 | 74% | 7 | 64% |
|  | >400 | 12 | 26% | 4 | 36% |
| Max seq sim to train | <0.5 | 13 | 28% | 4 | 36% |
|  | 0.5-0.6 | 18 | 38% | 3 | 27% |
|  | 0.6-0.7 | 11 | 23% | 1 | 9% |
|  | 0.7-0.8 | 5 | 11% | 0 | 0% |
|  | ≥0.8 | 0 | 0% | 3 | 27% |
| Neff of MSA | 0-10 | 8 | 17% | 1 | 9% |
|  | 10-100 | 23 | 49% | 5 | 45% |
|  | 100-1000 | 14 | 30% | 5 | 45% |
|  | 1000-10000 | 2 | 4% | 0 | 0% |

Table 2: rMSA databases used in RNA MSA search.

| Database | Temporal cutoffs |
| --- | --- |
| RNAcentral v20.0 | 2022/3/28 |
| Rfam v14.7 | 2021/12/9 |
| NCBI NT | 2022/10/3 |

## A.2 BASELINE MODELS

Table 3: Model versions used for baseline model inference.

| Model | Version | Training data cutoff | MSA used |
| --- | --- | --- | --- |
| Vfold Pipeline | VfoldPipeline-standalone-v2.0 (Download link requested in 2025/9/18) | unknown | No MSA needed |
| trRosettaRNA | Online server version updated in 2024/11/01, https://yanglab.qd.sdu.edu.cn/trRosettaRNA/ | unkown but in 2022/01-2024/11/01 | Same as our model |
| NuFold | Github version http://kiharalab.org/nufold/global_step145245.pt | 2022/2/28 | Same as our model |
| RhoFold+ | Github version, model_20221010_params.pt, no structure refinement RNA-FM trained on RNAcentral ≤v20.0 | 2022/4/13 ≤2022/3/28 (RNAcentral) | Same as our model |
| DRfold2 | Github version, model 0,1,2,8,9 in cfg_97, no structure refinement RCLM: epoch_67000 trained on RNAcentral v22.0 | 2023/12/31 2023/2/8 (RNAcentral) | No MSA needed |
| AlphaFold 3 | Online server https://alphafoldserver.com/ | 2021/9/30 | Searched by AlphaFold 3 Server |
| Protenix | Github version model_v0.2.0.pt | 2021/9/30 | Same as our model |

## A.3 ALGORITHMS

In this section, we present the major algorithms we modified (highlighted in yellow) in AlphaFold 3 Abramson et al. (2024). For the meaning of notations, please refer to the AlphaFold 3 paper.

**Algorithm 1** Main Inference Loop (Algorithm 1 in AlphaFold 3)

---

**def** MainInferenceLoop($\{\mathbf{f}^*\}$, rnalm , fusion_position , fusion , $N_{\text{cycle}} = 4, c_s = 384, c_z = 128$):

1: $\{s_i^{\text{inputs}}\} \leftarrow$ InputFeatureEmbedder($\{\mathbf{f}^*\}$)

2: $s_i^{\text{rnalm}} =$ GetRNAEmbeddings(rnalm, $\mathbf{f}^*$)
    *# Fusion position 1*

3: **if** fusion_position == s_inputs **then**

4:     $s_i^{\text{inputs}} \leftarrow$ fusion($s_i^{\text{inputs}}, s_i^{\text{rnalm}}$)

5: **end if**

6: $s_i^{\text{init}} \leftarrow$ LinearNoBias($s_i^{\text{inputs}}$)
    *# Fusion position 2*

7: **if** fusion_position == s_init **then**

8:     $s_i^{\text{init}} \leftarrow$ fusion($s_i^{\text{init}}, s_i^{\text{rnalm}}$)

9: **end if**

10: $z_{ij}^{\text{init}} \leftarrow$ LinearNoBias($s_i^{\text{inputs}}$) + LinearNoBias($s_j^{\text{inputs}}$)
    *# Fusion position 3*

11: **if** fusion_position == z_init **then**

12:     $z_{ij}^{\text{rnalm}} =$ LinearNoBias($s_i^{\text{rnalm}}$) + LinearNoBias($s_j^{\text{rnalm}}$)

13:     $z_{ij}^{\text{init}} \leftarrow$ fusion($z_{ij}^{\text{init}}, z_{ij}^{\text{rnalm}}$)

14: **end if**

15: $z_{ij}^{\text{init}} +=$ RelativePositionEncoding($\{\mathbf{f}^*\}$)

16: $z_{ij}^{\text{init}} +=$ LinearNoBias($f_{ij}^{\text{token\_bonds}}$)

17: $\{\hat{z}_{ij}\}, \{\hat{s}_i\} \leftarrow \mathbf{0}, \mathbf{0}$

18: **for** $c \in [1, \ldots, N_{\text{cycle}}]$ **do**

19:     $z_{ij} = z_{ij}^{\text{init}} +$ LinearNoBias(LayerNorm($\hat{z}_{ij}$))

20:     $\{z_{ij}\} =$ MsaModule($\{s_i^{\text{msa}}\}, \{z_{ij}\}, \{s_i^{\text{inputs}}\}$)

21:     $s_i = s_i^{\text{init}} +$ LinearNoBias(LayerNorm($\hat{s}_i$))

22:     $\{s_i\}, \{z_{ij}\} \leftarrow$ PairformerStack($\{s_i\}, \{z_{ij}\}$)

23:     $\{\hat{s}_i\}, \{\hat{z}_{ij}\} \leftarrow \{s_i\}, \{z_{ij}\}$

24: **end for**

25: $\{\vec{x}_i^{\text{pred}}\} \leftarrow$ SampleDiffusion($\{\mathbf{f}^*\}, \{s_i^{\text{inputs}}\}, \{s_i\}, \{z_{ij}\}$)

26: $p_{ij}^{\text{distogram}} \leftarrow$ DistogramHead($z_{ij}$)

27: **return** $\{\vec{x}_i^{\text{pred}}, p_{ij}^{\text{distogram}}\}$

---

---

**Algorithm 2** Diffusion Conditioning (Algorithm 21 in AlphaFold 3)

---

**def** DiffusionConditioning( $\hat{t}, \{\mathbf{f}^*\}, \{\mathbf{s}_i^{\text{inputs}}\}, \{\mathbf{s}_i^{\text{trunk}}\}, \{\mathbf{z}_{ij}^{\text{trunk}}\}, \{\mathbf{s}_i^{\text{rnalm}}\}, \{\mathbf{z}_{ij}^{\text{rnalm}}\}, \text{fusion\_position},$
fusion\_method, $\sigma_{\text{data}}, c_z = 128, c_s = 384)$ :

    *# Pair conditioning, fusion position 5*
1: **if** fusion\_position == z **then**
2:     **if** fusion\_method == add **then**
3:         $\mathbf{z}_{ij} = \text{concat}\left([\mathbf{z}_{ij}^{\text{trunk}} + \mathbf{z}_{ij}^{\text{rnalm}}, \text{RelativePositionEncoding}(\{\mathbf{f}^*\})]\right)$
4:     **else**
5:         $\mathbf{z}_{ij} = \text{concat}\left([\mathbf{z}_{ij}^{\text{trunk}}, \text{RelativePositionEncoding}(\{\mathbf{f}^*\}), \mathbf{z}_{ij}^{\text{rnalm}}]\right)$
6:     **end if**
7: **else**
8:     $\mathbf{z}_{ij} = \text{concat}\left([\mathbf{z}_{ij}^{\text{trunk}}, \text{RelativePositionEncoding}(\{\mathbf{f}^*\})]\right)$
9: **end if**                                       $\triangleright \mathbf{z}_{ij} \in \mathbb{R}^{c_z}$
10: $\mathbf{z}_{ij} \leftarrow \text{LinearNoBias}(\text{LayerNorm}(\mathbf{z}_{ij}))$
11: **for** $b \in [1, 2]$ **do**
12:     $\mathbf{z}_{ij} \mathrel{+}= \text{Transition}(\mathbf{z}_{ij}, n = 2)$
13: **end for**
    *# Single conditioning, fusion position 4*
14: **if** fusion\_position == s **then**
15:     **if** fusion\_method == add **then**
16:         $\mathbf{s}_i = \text{concat}\left([\mathbf{s}_i^{\text{trunk}} + \mathbf{s}_i^{\text{rnalm}}, \mathbf{s}_i^{\text{inputs}}]\right)$
17:     **else**
18:         $\mathbf{s}_i = \text{concat}\left([\mathbf{s}_i^{\text{trunk}}, \mathbf{s}_i^{\text{inputs}}, \mathbf{s}_i^{\text{rnalm}}]\right)$
19:     **end if**
20: **else**
21:     $\mathbf{s}_i = \text{concat}\left([\mathbf{s}_i^{\text{trunk}}, \mathbf{s}_i^{\text{inputs}}]\right)$
22: **end if**                                        $\triangleright \mathbf{s}_i \in \mathbb{R}^{c_s}$
23: $\mathbf{s}_i \leftarrow \text{LinearNoBias}(\text{LayerNorm}(\mathbf{s}_i))$
24: $\mathbf{n} = \text{FourierEmbedding}\left(\frac{1}{4}\log(\hat{t}/\sigma_{\text{data}}), 256\right)$
25: $\mathbf{s}_i \mathrel{+}= \text{LinearNoBias}(\text{LayerNorm}(\mathbf{n}))$
26: **for** $b \in [1, 2]$ **do**
27:     $\mathbf{s}_i \mathrel{+}= \text{Transition}(\mathbf{s}_i, n{=}2)$
28: **end for**
29: **return** $\{\mathbf{s}_i\}, \{\mathbf{z}_{ij}\}$

---

---

**Algorithm 3** Diffusion Module (Algorithm 20 in AlphaFold 3)

---

**def** DiffusionModule($\{\vec{\mathbf{x}}_l^{\text{noisy}}\}, \hat{t}, \{\mathbf{f}^*\}, \{\mathbf{s}_i^{\text{inputs}}\}, \{\mathbf{s}_i^{\text{trunk}}\}, \{\mathbf{z}_{ij}^{\text{trunk}}\}, \{\mathbf{s}_i^{\text{rnalm}}\}, \{\mathbf{z}_{ij}^{\text{rnalm}}\}$, fusion_position, fusion_method, $\sigma_{\text{data}} = 16, c_{\text{atom}} = 128, c_{\text{atompair}} = 16, c_{\text{token}} = 768$) :

    *# Conditioning*

1: $\{\mathbf{s}_i\}, \{\mathbf{z}_{ij}\} = \text{DiffusionConditioning}\left(\hat{t}, \{\mathbf{f}^*\}, \{\mathbf{s}_i^{\text{inputs}}\}, \{\mathbf{s}_i^{\text{trunk}}\}, \{\mathbf{z}_{ij}^{\text{trunk}}\}, \{\mathbf{s}_i^{\text{rnalm}}\}, \{\mathbf{z}_{ij}^{\text{rnalm}}\}, \right.$
$\left. \text{fusion\_position}, \text{fusion\_method}, \sigma_{\text{data}}\right)$

    *# Scale positions to dimensionless vectors with approximately unit variance.*

2: $\mathbf{r}_l^{\text{noisy}} = \vec{\mathbf{x}}_l^{\text{noisy}} / \sqrt{\hat{t}^2 + \sigma_{\text{data}}^2}$             $\triangleright \mathbf{r}_l^{\text{noisy}} \in \mathbb{R}^3$

    *# Sequence-local Atom Attention and aggregation to coarse-grained tokens*

3: $\{\mathbf{a}_i\}, \{\mathbf{q}_l^{\text{skip}}\}, \{\mathbf{c}_l^{\text{skip}}\}, \{\mathbf{p}_{lm}^{\text{skip}}\} = \text{AtomAttentionEncoder}\left(\{\mathbf{f}^*\}, \{\mathbf{r}_l^{\text{noisy}}\}, \{\mathbf{s}_i^{\text{trunk}}\}, \{\mathbf{z}_{ij}\}, c_{\text{atom}}, \right.$
$\left. c_{\text{atompair}}, c_{\text{token}}\right)$

                                          $\triangleright \mathbf{a}_i \in \mathbb{R}^{c_{\text{token}}}$

    *# Full self-attention on token level.*

4: $\mathbf{a}_i \mathrel{+}= \text{LinearNoBias}(\text{LayerNorm}(\mathbf{s}_i))$

5: $\{\mathbf{a}_i\} \leftarrow \text{DiffusionTransformer}\left(\{\mathbf{a}_i\}, \{\mathbf{s}_i\}, \{\mathbf{z}_{ij}\}, \beta_{ij} = 0, N_{\text{block}} = 24, \; N_{\text{head}} = 16\right)$

6: $\mathbf{a}_i \leftarrow \text{LayerNorm}(\mathbf{a}_i)$

    *# Broadcast token activations to atoms and run Sequence-local Atom Attention*

7: $\{\mathbf{r}_l^{\text{update}}\} = \text{AtomAttentionDecoder}\left(\{\mathbf{a}_i\}, \{\mathbf{q}_l^{\text{skip}}\}, \{\mathbf{c}_l^{\text{skip}}\}, \{\mathbf{p}_{lm}^{\text{skip}}\}\right)$

    *# Rescale updates to positions and combine with input positions*

8: $\vec{\mathbf{x}}_l^{\text{out}} = \sigma_{\text{data}}^2 / (\sigma_{\text{data}}^2 + \hat{t}^2) \cdot \vec{\mathbf{x}}_l^{\text{noisy}} \; + \; \sigma_{\text{data}} \cdot \hat{t} / \sqrt{\sigma_{\text{data}}^2 + \hat{t}^2} \cdot \mathbf{r}_l^{\text{update}}$

9: **return** $\{\vec{\mathbf{x}}_l^{\text{out}}\}$

---

## A.4 MODELS

### A.4.1 TRAINING HYPERPARAMETERS

We adopt a two-stage training approach, with the first stage warming up the adapters while keeping Protenix's weights frozen.For the cross-attention fusion adapter, we use a learning rate of 0.01; otherwise, we use a learning rate of 0.1.

Table 4: Training hyperparameters. ${rnalm_fusion_position}, ${rnalm_fusion_method}, ${use_msa} are variables subject to the experiments.

| | Training stage 1 | Training stage 2 |
|---|---|---|
| seed | 42 | 42 |
| data.train_sets | rna3db_all | rna3db_all |
| data.msa.enable_rna_msa | ${use_msa} | ${use_msa} |
| dtype | bf16 | bf16 |
| diffusion_batch_size | 48 | 48 |
| diffusion_chunk_size | 12 | 12 |
| iters_to_accumulate | 4 | 4 |
| train_crop_size | 384 | 384 |
| max_steps | 400 | 4000 |
| warmup_steps | 1 | 100 |
| learning_rate | 0.1/0.01 | 1e-3 |
| ema_decay | / | 0.999 |
| augment.fast_training | True | True |
| augment.freeze_backbone | True | False |
| augment.use_rnalm | True | True |
| augment.rnalm_name | aido_rna_650m | aido_rna_650m |
| augment.rnalm_fusion_position | ${rnalm_fusion_position} | ${rnalm_fusion_position} |
| augment.rnalm_fusion_method | ${rnalm_fusion_method} | ${rnalm_fusion_method} |

### A.4.2 INFERENCE HYPERPARAMETERS

Table 5: Inference hyperparameters. ${rnalm_fusion_position}, ${rnalm_fusion_method}, ${use_msa} are variables subject to the experiments.

|  | Description | Value |
|---|---|---|
| seeds | random seeds | 101 |
| model.N_cycle | number of recycles in Pairformer | 10 |
| use_msa | whether to use MSA or not | $use_msa |
| sample_diffusion.N_sample | number of structures for each target | 5 |
| sample_diffusion.N_step | number of diffusion steps | 200 |
| augment.use_rnalm | whether to use RNA LM or not | True |
| augment.rnalm_name | the RNA LM used | aido_rna_650m |
| augment.rnalm_fusion_position | RNA LM fusion position | ${rnalm_fusion_position} |
| augment.rnalm_fusion_method | RNA LM fusion method | ${rnalm_fusion_method} |

### A.5 ADDITIONAL RESULTS

### A.5.1 EFFECT OF LORA FINETUNING OF RNA LM FOR RNA STRUCTURE PREDICTION

In this section, we investigated beyond freezing the RNA LM in the RLM-aug Protenix. We performed LoRA fine-tuning (rank = 8) on the RNA LM within the RLM-aug Protenix (single conditioning, add) architecture, which introduces approximately 1.4 million additional trainable parameters. Since the LoRA adapters are newly initialized, we used a two-stage training schedule:

- Stage 1: train LoRA adapters and the projection matrix from LM to Protenix's representation space (lr=1e-3, frozen Protenix backbone, max_steps=800).

- Stage 2: jointly train adapters and Protenix (lr=1e-4).

All other hyperparameters were kept identical to those described in Appendix Table **??**. As shown in Appendix Table 6, the LoRA-finetuned variant exhibited a notable performance drop relative to the frozen model, suggesting that partial or full LM fine-tuning does not improve performance under current data limitations or may require extensive hyperparameter tuning to realize its potential.

Table 6: Effect of LoRA finetuning of RNA LM. Results are evaluated on RecentPBD-RNA.

|  | LM fintuning setting | TM-score | #success |
|---|---|---|---|
| RLM-aug Protenix w/ MSA (single cond., add) | Frozen | 0.438 | 14 |
|  | LoRA | 0.383 | 8 |
| RLM-aug Protenix w/o MSA (single cond., add) | Frozen | 0.389 | 12 |
|  | LoRA | 0.368 | 5 |

### A.5.2 CASE STUDY

For illustration, we visualize the predicted structures of our model alongside other methods on the test target 8SYK_A in Figure 1, where the ground truth structure from the PDB is shown in green.

### A.6 DATA AVAILABILITY

For the training data, it is publicly available in `https://github.com/marcellszi/rna3db/releases/tag/2024-12-04-full-release`. The MSAs for the training sequences are publicly available at folder `/MSA_v2` in `https://www.kaggle.com/competitions/stanford-rna-3d-folding/data`.

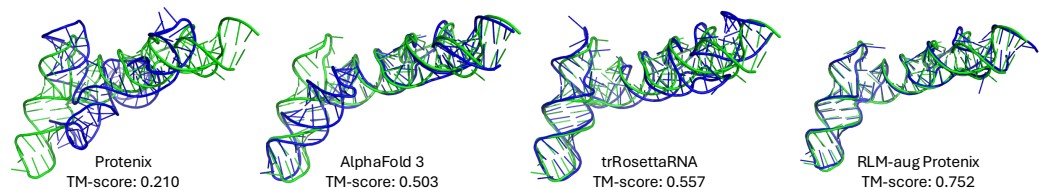

Figure 1: Visualization of PDB structure: 8SYK_A from RecentPDB-RNA. It is a synthetic RNA with 107 nucleotides, where the maximum sequence identity to the training set is 0.50. Green denotes ground truth structure, blue denotes predicted structure of the corresponding model.

For the detailed target list and the MSAs of RecentPDB-RNA and CASP16-RNA test sets, we will share them through our Github repository.

## A.7 CODE AVAILABILITY

Our code is largely based on Protenix https://github.com/bytedance/Protenix and AIDO.ModelGenerator https://github.com/genbio-ai/ModelGenerator. We will share our code and trained models on our GitHub repository.

