# OpenReview forum: "Accurate RNA 3D Structure Prediction via Language Model-Augmented AlphaFold 3"
_ICLR.cc/2026/Conference — Submitted to ICLR 2026_

### Official Review · Reviewer_Qgkw · 2025-10-26

**Soundness:** 1
**Presentation:** 1
**Contribution:** 1
**Rating:** 0
**Confidence:** 4

**Summary:**

The paper examines the integration of self-supervised RNA language models (RNA LMs) into AlphaFold 3 (AF3) to improve its RNA structure prediction capabilities. Different positions within the AF3 architecture and methods to fuse RNA LM embeddings were examined to achieve the best performance. Protenix, an open-source AF3 reproduction, which was trained on RNA data without using multiple sequence alignment (MSA), was fine-tuned on a custom, newly created RNA structure database, this time incorporating RNA LM embeddings and RNA MSAs. The paper reports a large performance variation based on the fusion position and methods, as well as including or not including MSAs. The authors report new state-of-the-art results on the custom RNA structure database, outperforming other deep learning RNA structure prediction methods, including the AF3 server. However, the performance boost mostly comes from augmenting the training dataset. On the CASP16 targets, the proposed RNA LM-augmented AF3 matches the best automated system trRosettaRNA, while still trailing behind the predicted structures of non-deep learning models with human interaction.

**Strengths:**

- Even though the idea of integrating self-supervised RNA LMs into RNA structure prediction models is not new and has already been investigated by replacing MSAs with them (DRfold2) or by augmenting MSA inputs (RhoFold+), examining the integration of an RNA LMs into AF3 is an interesting idea.
- The authors systematically evaluated different fusion methods and positions on the custom dataset, giving a clear idea of which strategy might improve the structure prediction performance.
- The authors promised to release data, code, and model weights to support reproducibility, which is important in the community dealing with a hard and delicate problem such as RNA structure prediction.
- Even though the proposed RNA LM-augmented AF3 model does not outperform the state-of-the-art automated system trRosettaRNA on the CASP16 targets, it boosts the performance of the server version of AF3. However, the way results are presented, it is not clear whether the boost comes from data augmentation, a different homolog search pipeline, or incorporating an RNA LM.

**Weaknesses:**

**Custom training and evaluation data:**
- For training data (finetuning data), the authors used the RNA3DB data released on 4 Dec 2024. This RNA3DB version takes into account all the existing RNA chains from the protein databank (PDB) released before 4 Dec 2024. For evaluation data, the authors created a custom dataset of RNA structures using only a time cut-off. The custom dataset was constructed of 67 samples from the PDB released between 4 Dec 2024 and 28 Apr 2025. The RNA structures in the evaluation dataset were not filtered by sequence and structure similarity to the training set. This means there is data leakage between the training and evaluation datasets. Other state-of-the-art tools make much more rigorous data splits. Why didn't you use the RNA3DB splitting strategy to make sure the RNAs in the custom evaluation dataset are structurally different from the training dataset? I suggest using the RNA3DB procedure, either including the newly released data or not, and using the *component #0* structures for the evaluation. That would provide a rigorous evaluation of the proposed method.

**Experimental setting and results:**
- From Table 1, it is clear that the biggest improvement from the original Protenix comes from data augmentation and possibly overfitting. The original Protenix used a time cut-off date of September 30, 2021 (same as AF3). From the table, it can be seen that adding only RNA LM embeddings either makes the performance worse or the performance in terms of TM-score is at most $ 0.003 $ higher, which is statistically insignificant. It is strained to say it is effective. Only after adding the MSAs, the results become better. Thus, it is not clear where the boost in performance comes from. Probably, it is the combination of both the MSA and RNA LM, but having only RNA LM is not effective. I think it would be beneficial to comment and explain why adding only RNA LM embeddings is not effective.
- (line 346) I have a problem with picking different models when reporting improvement in terms of percentage. For reported improvements, the authors picked interchangeably the worst of the two baseline models to report higher gains.
- In Table 2, the first column, it is not clear which versions of the concurrent models were used, their cut-off dates and whether the same MSAs as for the proposed model were employed. Additionally, even if the same MSAs were used, which were mined from the databases with temporal cutoffs before or 3 Oct 2022, the proposed model leveraged AIDO.RNA's embeddings, which were trained on the RNAcentral 24.0 dataset released on 25 Jun 2024. Reporting versions of the concurrent models used in comparison is the minimum I would expect. Also, I think it should be clearly stated and provided with the commands and inputs used for the inference of other tools. It would be fair to comment on the AIDO.RNA's pretraining dataset and its cutoff, and why wasn't the same cutoff date used for MSA search?
- In Table 2, the authors provided only the results of the proposed model finetuned on the augmented data, using MSA and RNA LM embeddings. It is crucial to see the effect of finetuning on the new dataset, which is not reported in Table 2. Additionally, since the benefits of using both MSA and RNA LM embeddings were not clear when evaluated on the custom evaluation dataset, I find it important to show their benefits separately for the CASP16 evaluation dataset. This way, it is not clear where the performance boost comes from; is it from data augmentation, using MSA, or employing RNA LM?
- In Table 3, the authors compared the proposed model with the finetuned Protenix; however, it is not clear whether this is the finetuned one with or without using MSA. This information is crucial since this way it is not clear whether the boost for low identity structures comes from using the MSA, RNA LM, or both. Please include this information to be more transparent.

**Minor comments**
- The reference list is often missing capitalization of letters in the paper titles and journal titles.

I recommend **rejecting** this paper.

The key reasons are:
1. Custom training and evaluation datasets are not able to test the generalization capabilities of the models for RNA structure prediction and are of little or no benefit.
2. The way the results are presented is poor and lacks transparency. Even if there is improvement over the AF3 model, the improvement does not outperform the state-of-the-art methods and it is definitely not clear where it comes from.

**Questions:**

- (line 241) It is not clear whether only 39% of the training sequences had available MSAs during the fine-tuning phase, or whether you additionally used rMSA for the rest of the sequences?
- Why didn't you use the RNA3DB splitting strategy to make sure the RNAs in the custom evaluation dataset are structurally different from the training dataset?
- In Table 2, please comment on the possiblilty of overfiting of your model on the custom dataset.
- Please could you provide the following:
  1. Report versions of the concurrent models used in comparison.
  2. Clearly state and provide the commands and inputs used for the inference of other tools.
- It would be fair to comment on the AIDO.RNA's pretraining dataset and its cutoff, and why wasn't the same cutoff date used for MSA search?
- In Table 2, for CASP16 results, could you provide results of the finetuned Protenix only using the augmented data, and the one finetuned using the augmented data and MSA?
- In Table 3, could you be more specific whether finetuned Protenix is the finetuned model with or without using MSA?

---

> ### Author Response · Authors · 2025-11-24
> **Response to  Reviewer Qgkw #1**
>
> **Response to Weakness 1 and Question 2**:
>
> Thank you for this comment and for the opportunity to clarify.
> 1. **Evaluation protocol.** Our curated RecentPDB-RNA dataset follows AF3’s temporal cutoff evaluation. We applied a sequence-similarity cutoff (<100%) and reported results stratified by similarity to the training set. To further improve rigor, we have filtered out sequences with ≥80% sequence similarity to the training data and updated all results on this stricter evaluation set in the revised manuscript.
> 2. **Why not RNA3DB splitting.** We agree that RNA3DB offers a stricter structure-based split. The two reasons that we did not use RNA3DB splitting are: (1) The RNA3DB split is fully based on structure similarity and ignores temporal separation. Its test set contains structures already seen during Protenix training, which would itself cause data leakage; (2) Predicting structurally dissimilar RNAs is extremely challenging. As discussed in the recent Das Lab report (Assessment of nucleic acid structure prediction in CASP16, bioRxiv 2025), “3D accuracy generally appeared to depend on the availability of closely related 3D structures”. Given that performance on the temporal split remains challenging at this stage, directly targeting out-of-distribution generalization would be premature.
> 3. **On Component #0.** As noted in the RNA3DB paper, Component #0 includes 376 chains with no detectable Rfam homology, often synthetic constructs, mRNA fragments, or short structural segments. The authors themselves state that “RNA3DB gives Component #0 the option to ignore this altogether.” For completeness, we will include experiments using the RNA3DB split (including Component #0) in the revision to evaluate our model’s out-of-distribution generalization.
>
>
> **Response to Weakness 2.1**:
>
> Thank you for this insightful comment. We performed a 2×2 ablation study examining the effects of MSA and RNA LM embeddings when fine-tuning Protenix on the same dataset and evaluating on RecentPDB-RNA (with an 80% sequence similarity cutoff). As shown in **Figure 2** in our revised manuscript, using either MSA alone or RNA LM alone slightly decreases performance relative to the fine-tuned Protenix baseline without MSA, whereas combining both MSA and RNA LM embeddings achieves the best overall results, indicating strong complementarity. We hypothesize that this synergy arises because both MSA and RNA LM embeddings encode evolutionary information, but in distinct feature spaces. Protenix is originally optimized to leverage MSA-based features (from protein training), so introducing RNA LM embeddings alone without MSA makes it difficult for the model to align these heterogeneous representations. When used together, however, the LM embeddings regularize and enrich the MSA-derived signals, leading to improved generalization. We will include detailed analyses of this interaction in the revised version.
>
>
> **Response to Weakness 2.2**:
>
> Thank you for pointing this out. We have clarified that the baseline used for reporting improvements is the Finetuned Protenix with MSA, which is the appropriate reference for our model that is also finetuned with MSA. The two reported improvements (7% and 16%) correspond, respectively, to the relative gains in TM-score and #success over this same baseline.

---

> ### Author Response · Authors · 2025-11-24
> **Response to Reviewer Qgkw #2**
>
> **Response to Weakness 2.3, Question 4, and Question 5:**
>
> Thank you for this helpful comment and the opportunity to clarify.
>
> 1. **Model versions and MSA usage.** We have now specified the exact versions of all baseline models in **Appendix Table 3** in the revised manuscript. For models that utilize MSAs, we used the same MSAs generated by our search pipeline, except for the AlphaFold 3 Server, which does not allow user-supplied MSAs. We will include the model versions in the revised paper to ensure full transparency.
> 2. **Inference settings.**
>     - Vfold-Pipeline: We followed the configuration in VfoldPipeline-standalone-manual.pdf, with `if_PK = No` and `cluster_num = 5`.
>     - RhoFold+, NuFold, and Protenix: Inference followed their official GitHub instructions. No structure refinement for RhoFold+.
>     - DRfold2: The official inference script was prohibitively slow and insufficiently documented (80 models without clear distinctions). Instead, we adopted the settings from the top-performing public notebook in the Stanford RNA 3D Folding Kaggle Competition, which has been optimized for RNA 3D structure accuracy. No structure refinement.
>     - trRosettaRNA: We uploaded our test sequences and the MSAs we generated to the public server.
>     - AlphaFold 3: We uploaded test sequences directly and used the server's internal MSA pipeline for structure prediction.
>     We will release the notebooks we used for baseline models, together with our code in revision.
> 3. **AIDO.RNA pretraining and MSA cutoff.**
> AIDO.RNA was pretrained on RNAcentral 24.0 (released 7 March 2024). The MSA searches were performed using rMSA, which automatically accesses its most recent database version at the time of inference (as detailed in Appendix Table 2). Because rMSA does not allow customization of the underlying database, we used the default version it provides. This results in a difference between the cutoff date of the AIDO.RNA pretraining data and that of the MSA search database.
>
> **Response to Weakness 2.4 and Question 6**:
>
> Thank you for pointing this out. We have updated **Table 2** to include Finetuned Protenix with MSA as a reference and added results for two additional RLM-aug Protenix variants trained with MSA. On the 11 CASP16-RNA targets, all three RLM-aug Protenix models achieve higher or slightly higher TM-scores than Finetuned Protenix with MSA. We note that the performance trends for our models on CASP16 differ from those observed on RecentPDB-RNA. Given the very limited size of the CASP16-RNA test set, we consider it inappropriate to draw strong conclusions about generalization based solely on this dataset. We fully agree that generalization is crucial and are currently curating additional, recently released RNA structures from the PDB to expand our evaluation.
>
> **Response to Weakness 2.5 and Question 7**:
>
> We apologize for the confusion. The Finetuned Protenix in **Table 3** is trained with MSA. We have included the information in our revision.
>
> **Response to Minor comments**:
>
> Thank you for catching this. We have carefully reviewed the reference formatting and corrected the capitalization of journal titles. Regarding paper titles, the ICLR reference style, as observed in previous ICLR publications, capitalizes only the first letter of the first word, and we have ensured our references follow this convention.
>
> **Response to Question 1**:
>
> Thank you for the opportunity to clarify. The MSAs used during finetuning were precomputed with rMSA prior to training. During fine-tuning, we simply retrieved these precomputed MSAs for the corresponding sequences. We did not perform additional rMSA searches for sequences without available MSAs.
>
> **Response to Question 3**:
>
> Thank you for the opportunity to clarify. The potential overfitting issue arises from test sequences with more than 80% similarity to the training set. To address this, we have filtered out all sequences with ≥80% similarity and updated **Table 2** accordingly in the revised manuscript. After this correction, our RLM-aug Protenix with MSA (single_conditioning, add) continues to achieve the best overall performance, indicating that the observed improvements are not due to overfitting.

---

> > ### Comment · Reviewer_Qgkw · 2025-11-25
> > **Response to authors**
> >
> > Dear authors,
> >
> > Thank you for the revision of your paper and the clarifications that improve its presentation.
> >
> > I still have concerns regarding the paper that I will list below.
> >
> > **Response to Weakness 1 and Question 2:**
> > 1. I appreciate the improvement of rigor and setting the sequence similarity threshold to 80%. However, I still want to stress the importance of structure similarity, which is strictly required in high-quality journals where any overlaps between training and testing datasets must be explicitly justified in the manuscript.
> > 2. You stated that "The RNA3DB split is fully based on structure similarity and ignores temporal separation. Its test set contains structures already seen during Protenix training, which would itself cause data leakage." However, it is possible to incorporate the temporal separation during the curation, e.g., see [1].
> >
> > **Response to Weakness 2.1:**
> >
> > This still remains my biggest weakness in the paper. From Tables 1, 2 and 3, it is unclear where the performance boost originates. For sure, the biggest improvement from the original Protenix comes from data augmentation. However, adding RNA LM and MSA results in mixed performance, depending on where the RNA LM features are incorporated. After reading the paper, there is no clear answer or recipe on how to incorporate RNA LM in Protenix and whether this will lead to improvement or degradation in performance for a custom set of RNAs.
> >
> > **Response to Weakness 2.3, Question 4, and Question 5:**
> > 1. Thank you for listing the exact versions of all baseline models. Since the models were trained using different MSA tools, I think it would be fair to comment on the possible impact on the final performance when using the same tool for all the models.
> > 2. Why didn't you use structure refinement for RhoFold+ and DRfold2? It's a part of their standard prediction protocol.
> > 3. RNA LMs are really strong at modeling homology. Your model uses AIDO.RNA whose pretraining data cutoff date was 7 March 2024. MSAs were mined from the databases with temporal cutoffs before or 3 Oct 2022. Could you comment on the possible leveraging of RNA LM's extended dataset to achieve better results compared to other models that used only MSAs? Wouldn't it be more fair to use RNA LMs pretrained on data with an earlier cutoff that match MSA temporal cutoffs, e.g., RNA-FM [2].
> >
> > I appreciate the effort put in by the authors to improve the work, however I still stick to my previous decision.
> >
> > ---
> > [1] Martinović, I., Vlašić, T., Li, Y., Hooi, B., Zhang, Y., & Šikić, M. (2024). A Comparative Review of Deep Learning Methods for RNA Tertiary Structure Prediction. bioRxiv, 2024-11.
> >
> > [2] Chen, J., Hu, Z., Sun, S., Tan, Q., Wang, Y., Yu, Q., ... & Li, Y. (2022). Interpretable RNA foundation model from unannotated data for highly accurate RNA structure and function predictions. arXiv preprint arXiv:2204.00300.

---

> > > ### Author Response · Authors · 2025-11-29
> > > **Response to Reviewer Qgkw #3**
> > >
> > > Thank you for your prompt response.
> > >
> > > **Response to Weakness 1 and Question 2:**
> > >
> > > 1. Thank you for this comment. We acknowledge that structural dissimilarity provides the most stringent test of model generalization. However, as noted in the literature and demonstrated in several recent studies—such as trRosettaRNA (Nature Communications, 2023), RhoFold+(Nature Methods, 2024), and NuFold (Nature Communications, 2025)—**the use of temporal and sequence similarity cutoffs remains a standard and widely accepted practice for evaluating RNA 3D prediction models.** In the reference work you mentioned ([1]), the authors also adopted sequence similarity thresholds (e.g., Dataset3 with 90% cutoff) to assess generalization, consistent with the practices of other high-impact studies. While we fully agree that structure-based filtering represents the next level of rigor, we view our current benchmark—combining a temporal cutoff and sequence identity threshold of 80%—as an important step toward that goal. Our plan is to progressively incorporate structure dissimilarity filtering in future benchmarks as larger, higher-quality RNA structure datasets become available.
> > >
> > > 2. Thank you for the suggestion. We will incorporate temporal separation into the RNA3DB structure-dissimilar split and include these results in the revised manuscript as an additional test of generalization. It is important to note, however, that—as shown in [1]—performance on sequence-dissimilar targets (Dataset 3) is already low for all strong models, including AF3 and trRosettaRNA (Figure 2F, TM-score < 0.3), indicating that **the field still struggles with sequence-dissimilar generalization.** Performance on structure-dissimilar targets (Dataset 4) is even lower across all methods (Figure 5B), highlighting that this evaluation setting remains substantially more challenging than what current models can reliably handle.
> > >
> > > **Response to Weakness 2.1:**
> > >
> > > Thank you for raising this important point. We agree that clearly identifying where the performance gains come from is essential, and we have revised the manuscript to clarify this.
> > >
> > > First, Table 1 (revised manuscript) shows that RLM-aug Protenix (single-conditioning, add) with MSA achieves significantly higher TM-scores than both Protenix baselines and other fusion variants (one-sided paired t-test, α = 0.05). This provides a clear and actionable recipe:
> > >
> > > **→ Use single-conditioning + additive fusion, together with MSA during fine-tuning.**
> > >
> > > Second, we now explicitly quantify the contributions of data & MSA and RNA LM in the following table and also Table 3 in the revised manuscript. Across all targets, data + MSA account for ~66% of the total performance improvement, while RNA LM contributes ~34%. When grouping test sequences by sequence similarity to the training set, we observe a clear trend:
> > > * For high-similarity targets, gains come mostly from data & MSA;
> > > * For low-similarity targets, the RNA LM contribution becomes dominant.
> > >
> > > |                                  | All (47) | 0.7–0.8 (5) | 0.6–0.7 (11) | 0.5–0.6 (18) | <0.5 (13) |
> > > |--------------------------------------------|----------|-------------|--------------|--------------|------------|
> > > | [1] Protenix                                   | 0.325    | 0.266       | 0.458        | 0.333        | 0.225      |
> > > | [2] Finetuned Protenix w/ MSA                  | 0.399    | 0.539       | 0.592        | 0.356        | 0.242      |
> > > | [3] RLM-aug Protenix w/ MSA (single cond., add)| 0.438    | 0.609       | 0.619        | 0.377        | 0.303      |
> > > | Improvement from Data & MSA ([2]-[1])          | 0.074    | 0.273       | 0.134        | 0.023        | 0.017      |
> > > | Improvement from RNA LM ([3]-[2])              | 0.039    | 0.070       | 0.028        | 0.021        | 0.061      |
> > > | Contribution of Data & MSA ( ([2]–[1])/([3]–[1]) ) | 66%      | 80%         | 83%          | 52%          | 22%        |
> > > | Contribution of RNA LM ( ([3]–[2])/([3]–[1]) )     | 34%      | 20%         | 17%          | 48%          | 78%        |
> > >
> > > **Table 1.** TM-score on RecentPDB-RNA by maximum sequence similarity to the training set.
> > >
> > > Third,  as shown in Figure 4 (revised manuscript),  RLM-aug Protenix w/ MSA (single cond., add) improves performance on long RNAs (nt > 400)—a particularly challenging regime in RNA 3D structure prediction.
> > >
> > > Together, these analyses clarify the role of each component and provide practitioners with concrete guidance on when and how RNA LM integration is most beneficial.

---

> > > ### Author Response · Authors · 2025-11-29
> > > **Response to Reviewer Qgkw #4**
> > >
> > > **Response to Weakness 2.3, Question 4, and Question 5:**
> > >
> > > 1. Thank you for this comment. We agree that using a consistent MSA source is important for fair comparison. In our evaluation, **all models that can accept external MSAs use the same rMSA-generated MSAs**. However, Vfold-Pipeline and DRfold2 are inherently MSA-free, and the AlphaFold 3 server does not allow user-provided MSAs. Given these constraints, a setting in which all models use the same MSA tool is not feasible.
> > >
> > > 2. Thank you for this question. Both RhoFold+ and DRfold2 provide an optional AMBER relaxation step implemented via OpenMM (https://openmm.org/), which performs post-processing on the predicted PDB structure using the AMBER molecular mechanics force field through energy minimization. The same refinement can also be applied to predictions of AlphaFold 3, Protenix, and our own models. To ensure a fair and controlled comparison, we intentionally excluded the structure refinement so that the reported performance reflects the capabilities of the prediction models themselves rather than differences introduced by post-processing. In addition, as noted in the RhoFold+ paper, the relaxation does not improve accuracy in terms of TM-score or r.m.s.d.; rather, its primary effect is to eliminate distracting stereochemical violations without compromising accuracy.
> > >
> > > 3. Thank you for raising this point.
> > >       *  First, we do not assume—and are not aware of evidence in the literature—that current RNA LMs are “strong at modeling homology” in the same way that deep MSAs capture evolutionary relationships. RNA LM embeddings do contain useful contextual and evolutionary signals, but their homology modeling capability has not been shown to replace or match MSA search.
> > >       * Second, even if we hypothetically assume RNA LMs could capture homology, the incremental sequence data available to AIDO.RNA beyond the MSA cutoff is very small relative to the databases used for MSA generation. AIDO.RNA is pretrained on RNAcentral v24.0 (~30B nt), whereas rMSA uses multiple large databases—including RNAcentral v20.0, Rfam, and especially NCBI nt, which is an order of magnitude larger than RNAcentral (as reported in the rMSA paper). Thus, the difference between RNAcentral v24.0 and v20.0 is negligible compared to the scale of the nt database, which AIDO.RNA does not use. For this reason, we believe the LM’s extended sequence coverage does not provide a meaningful advantage over other models that use only MSA.
> > >      * Finally, our structure-data temporal cutoff is December 4, 2024, so using an RNA LM pretrained on sequence data released before that date is consistent with our overall data-split strategy. We used AIDO.RNA because, at the time of writing, it is the largest available RNA LM and consistently outperforms RNA-FM across multiple downstream RNA tasks (secondary structure, function prediction), making it the strongest choice for evaluating fusion strategies.

---

### Official Review · Reviewer_JaFT · 2025-10-29

**Soundness:** 3
**Presentation:** 2
**Contribution:** 2
**Rating:** 6
**Confidence:** 3

**Summary:**

This paper explores methods to enhance RNA 3D structure prediction in AlphaFold 3 by incorporating embeddings from a pretrained RNA language model. The authors operate on the premise that the protein-centric training of AF3 leaves its RNA-specific representations underdeveloped. This model is benchmarked on the RecentPDB-RNA dataset, where it shows improved average TM-scores compared to the baseline AF3, and on 11 CASP16-RNA targets, where its performance is on par with trRosettaRNA.

**Strengths:**

1. The paper tackles a relevant and challenging problem in structural biology, and the hypothesis that RNA-specific LMs can help AF3 is well-motivated.

2. The experimental design is systematic. The controlled study of fusion positions and methods provides a clear ablation, which is a useful, if straightforward, contribution.

3. The analysis in Section 6.3, which breaks down performance by sequence length and identity, is helpful. It correctly identifies that the LM provides the most benefit in data-sparse scenarios (low identity, long sequences), which supports the original hypothesis.

**Weaknesses:**

1. The primary contribution of this work is empirical, not methodological. The paper does not propose any new fusion techniques but rather applies existing ones. The main takeaway is "additive fusion at position X works best," which is a useful engineering finding but lacks deeper algorithmic insight.

2. The performance claims seem somewhat inflated when considering the full picture. While the 21% relative gain on RecentPDB-RNA is notable, the model only matches the existing trRosettaRNA server on the CASP16 benchmark. This lack of a clear win on the CASP set calls into question how broadly the SOTA claims generalize.

3. The practical utility of the resulting model is questionable. The authors state that the confidence head was not fine-tuned. This is a significant omission, as it means the model cannot provide a reliable estimate of its own accuracy, a feature that is essential for real-world use.

4. The exploration of methods feels incomplete. For instance, the RNA LM is kept frozen. It is a missed opportunity not to investigate at least partial finetuning of the LM, which might have yielded different or superior results.

**Questions:**

1. The paper concludes that fusing at pair representations is ineffective. Was this conclusion based solely on the outer-sum method for generating $z^{rnalm}$?

2. The paper does not provide an analysis of why simple additive fusion outperformed the more expressive cross-attention method. Is this simply because the RNA3DB training set is too small to properly train the attention adapter, or is there a more fundamental reason?

3. The results regarding MSAs in Table 1 are contradictory and unexplained. The Finetuned Protenix baseline actually performs slightly worse when MSAs are added (0.450 vs 0.441). In contrast, the RLM-aug Protenix model gets a significant boost from MSAs (0.431 vs 0.472). This suggests a non-trivial interaction between the LM features and the MSA features, which the paper fails to investigate. Can you explain this discrepancy?

---

> ### Author Response · Authors · 2025-11-25
> **Response to Reviewer JaFT #1**
>
> Thank you for your insightful comments and questions.
>
> **Response to Weakness 2:**
>
> Thank you for this comment. On RecentPBD-RNA, a key concern about potential overfitting comes from the sequence similarity control. To improve rigor, we have filtered out sequences with ≥80% sequence similarity to the training data and updated results accordingly in the revised manuscript. After this correction, our RLM-aug Protenix with MSA (single conditioning, add) continues to achieve the best overall performance, indicating that the observed improvements are not due to overfitting.  On CASP16-RNA, although our model does not significantly outperform trRosettaRNA, it achieves comparable accuracy. We note that the test set is very small (n=11) to draw strong conclusions, since small datasets can easily lead to large variations. To further assess generalization, we are curating newly released PDB RNA structures as an additional evaluation benchmark and will include them in the revised manuscript.
>
> **Response to Weakness 3:**
>
> Thank you for this insightful comment. We will add a new stage of finetuning, which finetunes the Confidence Head while keeping the other weights frozen in the revised manuscript.
>
> **Response to Weakness 4:**
>
> Thank you for the suggestion. We chose to freeze the RNA LM because the fine-tuning dataset is relatively small, and updating all LM parameters would likely lead to overfitting. The RNA LM (650M parameters) is even larger than the Protenix backbone (386M parameters), making full joint training impractical with the available data. Similar design choices are adopted in ESMFold (Lin et al., Science, 2023), RhoFold+, and DRfold2, where pretrained LMs are also kept frozen during structure model training.
> To further investigate, we performed LoRA fine-tuning (rank = 8) on the LM within the RLM-aug Protenix (single conditioning, add) architecture, introducing approximately 1.4 M additional trainable parameters. Since the LoRA adapters are newly initialized, we used a two-stage training schedule:
> * Stage 1: train LoRA adapters and the projection matrix from LM to Protenix’s representation space (lr = 1e−3, frozen Protenix backbone, max_steps=800).
> * Stage 2: jointly train adapters and Protenix (lr = 1e−4).
> All other hyperparameters were kept identical to those described in the Appendix. As shown in the following Table, the LoRA-finetuned variant exhibited a notable performance drop relative to the frozen model, suggesting that partial or full LM fine-tuning does not improve performance under current data limitations, or may require extensive hyperparameter tuning to realize its potential.
>
> | Model                                      | LM Fine-tuning Setting | TM-score | #Success |
> |--------------------------------------------|------------------------|-----------|-----------|
> | RLM-aug Protenix w/ MSA (single cond., add)  | Frozen                | 0.438     | 14        |
> | RLM-aug Protenix w/ MSA (single cond., add)  | LoRA                  | 0.383     | 8         |
> | RLM-aug Protenix w/o MSA (single cond., add) | Frozen                | 0.389     | 12        |
> | RLM-aug Protenix w/ MSA (single cond., add)  |  LoRA                  | 0.368     | 5         |
> **Table 1.** Effect of LM fine-tuning on model performance on RecentPDB-RNA.
>
> **Response to Question 1:**
>
> Thank you for this question. The claim that fusing at the pair representation is ineffective was solely based on the outer sum of RNA LM embeddings. We apologize for rushing to claim this, since the reason could be that the outer-sum adapter weights are not well-trained, or the input information is not structurally informative enough. We had preliminary results that show fusing secondary structure prediction into the pair representation is helpful.
>
> **Response to Question 2:**
>
> Thank you for this question. The main reasons are: (1) the limited size of RNA3DB, which makes it difficult to train the higher-capacity cross-attention adapter effectively; and (2) cross-attention’s larger parameter count, which requires more data and iterations to converge, whereas additive fusion is simpler and more stable under limited data.

---

> ### Author Response · Authors · 2025-11-25
> **Response to Reviewer JaFT #2**
>
> **Response to Question 3:**
>
> Thank you for this insightful question.
>
> We performed a 2×2 ablation study examining the effects of MSA and RNA LM embeddings when fine-tuning Protenix on the same dataset and evaluating on RecentPDB-RNA. As shown in **Figure 2** in the revised manuscript, using either MSA alone or RNA LM alone slightly decreases performance relative to the finetuned Protenix baseline without MSA, whereas combining both MSA and RNA LM embeddings achieves the best overall results, indicating strong complementarity.
>
> We hypothesize that this synergy arises because both MSA and RNA LM embeddings encode evolutionary information, but in distinct feature spaces. Protenix is originally optimized to leverage MSA-based features (from protein training), so introducing RNA LM embeddings alone without MSA makes it difficult for the model to align these heterogeneous representations. When used together, however, the LM embeddings regularize and enrich the MSA-derived signals, leading to improved generalization.

---

### Official Review · Reviewer_QsMV · 2025-10-29

**Soundness:** 3
**Presentation:** 4
**Contribution:** 3
**Rating:** 6
**Confidence:** 4

**Summary:**

The paper analyses the incorporation of RNA language model embeddings into AlphaFold 3 (or Protenix as an open-source reproduction of AF3). The authors identify five different positions in the AF3 architecture to enhance it with LM embeddings, and study three different strategies for embedding fusion (add, concat, attention-based) using RNA3DB. The best performing model, RLM-aug Proteinx, is further evaluated on 67 recent PDB samples, as well as 11 CASP16 RNA targets. Here, RLM-aug Protenix shows substantial performance improvements with SOTA results compared to state-of-the-art 3D folding engines on the recent PDB dataset and competitive performance with the best automated method for the CASP16 data. The authors further analyze the model performance across different sequence lengths and wrt. sequence identity between training and test samples.

**Strengths:**

- The authors provide a solid and well controlled experimental setup for evaluation of 3D predictions using their different embedding strategies.
- The model achieves SOTA performance on a PDB dataset of recent samples and general strong performance for RNA 3D folding.
- The incorporation of RNA-LM embeddings from massive self-supervised sequence learning into an existing SOTA 3D folding engine is promising, especially since the proposed model clearly outperforms existing approaches like RhoFold+ and DRFold2.
- Improving 3D RNA predictions is a challenging and important task.

**Weaknesses:**

1. While the authors thoroughly evaluate 3D structure predictions, there is nearly no assessment of the embedding quality (except for final 3D structure prediction quality). I’m not super familiar with embedding analysis of RNA-LMs, but a simple experiment could for example be to score embedding quality, e.g. using IsoScore [1] or other methods as recently done for DNA embeddings [2]. Then one could analyze if embedding “quality” correlates with structure quality.
2. Another aspect is the usage of MSA. If I understand correctly, MSA was not available for all datapoints. Analyzing predictions with and without available MSA and potentially also relating to embedding information could provide further useful insights.
3. The authors use only a single RNA-LM for their evaluation. The only comment on the choice is that it is a “strong transformer-based encoder only LM”. Since the paper ‘only’ assesses different ways to incorporate RNA-LM embeddings into Protenix, different LMs could have been tested, although this would likely require effort to match embedding dimensions.

[1] Rudman, W., Gillman, N., Rayne, T., & Eickhoff, C. (2021). IsoScore: Measuring the uniformity of embedding space utilization. arXiv preprint arXiv:2108.07344.

[2] Awasthi, R., Mend Mend Arachchige, G. S., & Zhu, X. (2025). Unsupervised evaluation of pre-trained DNA language model embeddings. BMC genomics, 26(1), 710.

**Questions:**

1. In table 1, do the authors have any rationale why performance for all methods seems to decrease when using MSA, except for the final RLM-aug Protenix (and success rate for FT Protenix)?
2. Since the authors mention RNA flexibility as a challenge for 3D structure prediction, does the incorporation of RNA-LM features influence the sampling behavior of the model? While sampling in AF3 (and I expect it to be similar for Protenix) does not provide true conformation ensembles, it would still be interesting to see if the structure distribution of the five samples per test target changes with embeddings.
3. I can imagine that the embedding quality of the LM is higher if the model already knows similar sequences from the training data. While done for training data for 3D predictions, how does the training data of the LM relate to structure prediction quality?
4. Since the performance drops quite substantially for longer RNAs, could training the LM on more longer sequences solve the problem? Or more general, does the length distribution of RNAcentral correlate with the observed performance across different lengths?
5. The LM was trained on ncRNAs from RNAcentral only, how about performance on coding RNAs? How does performance look like for different RNA types? Are there datapoints available to analyze this kind of questions? Can this also be related to the embedding quality?

---

> ### Author Response · Authors · 2025-11-25
> **Response to Reviewer QsMV #1**
>
> We sincerely appreciate your constructive and thoughtful comments and questions, which provide valuable guidance and inspiration for our next steps.
>
> **Response to Weakness 1:**
>
> Thank you for this helpful suggestion. We will investigate the embedding space to see how it affects the structure quality in the revised manuscript.
>
> **Response to Weakness 2:**
>
> Thank you for this insightful perspective.
>
> We plot the TM-score improvement obtained by inference with MSA compared to inference without MSA in **Figure 3** in the revised manuscript. For the two Protenix models without RNA LM, incorporating MSA generally did not improve performance, especially for low-Neff (number of effective sequences in MSA) sequences. In contrast, the three models trained with RNA LM + MSA showed a consistent positive trend: as Neff increased, the performance gain from MSA became more pronounced. Notably, the RLM-aug Protenix (single-conditioning, add) variant exhibited the strongest ability to leverage MSA information, achieving the largest improvements with higher Neff. These results demonstrate that integrating an RNA LM enhances the model’s capacity to exploit evolutionary information from MSAs during inference.
>
> **Response to Weakness 3:**
>
> Thank you for pointing this out. We will include different RNA LMs in our revision.
>
> **Response to Question 1:**
>
> Thank you for this question. We think there are several factors contributing to the observed performance trend:
>
> - MSA quality: The RNA MSAs in the test set are relatively shallow—17% of samples have Neff < 10, and 66% have Neff < 100 (**Appendix Table 1**, revised manuscript). As Neff increases, model performance also improves.
> - Limited MSA utilization: The Finetuned Protenix w/ MSA model struggles to effectively use RNA MSA information during inference, even as Neff grows (**Figure 3**, revised manuscript).
> - LM-assisted MSA usage: When RNA LM embeddings are introduced during fine-tuning, the models learn to better exploit MSA information at inference across all three variants, with the (single-conditioning, add) configuration showing the largest gains (**Figure 3**, revised manuscript).
> - Fusion design: Among the three fusion strategies, single-conditioning fusion best preserves the original MSA and Pairformer information flow, maintaining structural priors while integrating LM embeddings at a later stage for improved conditioning.
>
> **Response to Question 2:**
>
> Thank you for this insightful question. To evaluate whether RNA LM features affect sampling behavior, we analyzed five sampled structures per test target and computed:
> * the standard deviation (STD) of TM-scores against ground truth, reflecting diversity
> * the mean similarity (TM-score) between samples, reflecting self-consistency among predictions.
>
> As shown in the following table, when comparing RLM-aug Protenix w/ MSA (single_cond., add)  with Finetuned Protenix w/ MSA, the similarity between samples is very close, while the STD of sample TM-scores shows a difference. It probably indicates that the incorporation of LM slightly increases the sampling diversity.
> | Model                                               | STD of sample TM-scores | Similarity between samples | TM-score |
> |-----------------------------------------------------|--------------------------|-----------------------------|----------|
> | AlphaFold 3                                         | 0.019                   | 0.568                       | 0.358    |
> | Protenix                                            | 0.022                   | 0.498                       | 0.325    |
> | Finetuned Protenix w/ MSA                           | 0.022                   | 0.574                       | 0.399    |
> | RLM-aug Protenix w/ MSA (init_single_rep, add)      | 0.022                   | 0.574                       | 0.409    |
> | RLM-aug Protenix w/ MSA (init_single_rep, cross_atten.) | 0.020               | 0.583                       | 0.409    |
> | RLM-aug Protenix w/ MSA (single_cond., add)         | 0.031                   | 0.573                       | 0.438    |
> **Table 1.** Sampling consistency analysis on RecentPBD-RNA.

---

> ### Author Response · Authors · 2025-11-25
> **Response to Reviewer QsMV #2**
>
> **Response to Question 3:**
>
> Thank you for this question. Self-supervised pretraining on large RNA sequence corpora enables language models (LMs) to learn evolutionary and contextual dependencies that are highly informative for structure prediction. In the protein domain, studies such as “Transformer Protein Language Models Are Unsupervised Structure Learners” (Rao et al., 2021, ICLR) have shown that pretrained LMs implicitly capture structural information even without explicit 3D supervision. We believe similar principles apply in the RNA domain, where evolutionary correlations encoded in sequence patterns can translate into useful structural priors.
>
> Furthermore, SimpleFold (Wang et al., 2025, arXiv) demonstrated that explicit structure-inspired modules (e.g., triangle attention or pair representation biases) are not strictly necessary when high-quality LM embeddings are available. This indicates that the structural and evolutionary information captured by the LM can be effectively utilized by simpler downstream architectures—an observation consistent with our findings that RNA LM embeddings enhance the use of MSA information during inference and structure prediction quality.
>
> **Response to Question 4:**
>
> Thank you for this question. Predicting large RNAs remains inherently difficult due to their combinatorial structural complexity, long-range dependencies, sparse evolutionary signals, and the limited availability of experimental structures—only about 29% of our training data correspond to large RNAs. These challenges make modeling long sequences substantially harder than short ones.
>
> The RNA LM we used was pretrained with a maximum sequence length of 1024 on RNAcentral that has an average of 728 nucleotides, which already covers all sequences in our test sets (maximum length <900 nt). Thus, extending LM pretraining to longer sequences is unlikely to resolve this issue. Instead, the bottleneck lies in the structure model, which must be trained on more long-RNA examples to learn complex tertiary dependencies.
>
> However, this is currently constrained by data scarcity and computational cost—the Pairformer architecture scales as O(L³) with sequence length, making long-sequence training computationally demanding. For instance, AlphaFold 3 supports training with sequences up to 768 nt, while Protenix handles up to 640 nt on NVIDIA A/H100 (80 GB) GPUs. Addressing long-RNA prediction will thus require both larger, higher-quality datasets and scalable architectures optimized for efficiency.
>
> **Response to Question 5:**
>
> Thank you for this question. The RNA 3D structures available in the PDB are predominantly ncRNAs, as structural studies have focused on molecules such as rRNAs, tRNAs, riboswitches, and ribozymes, whose biological functions are closely tied to their 3D conformations. In contrast, mRNAs function mainly as templates for protein synthesis, giving structural biologists less incentive to determine their 3D structures. As a result, our current training data include very few coding RNAs, and the test sets contain none, making it infeasible to perform a meaningful comparison of model performance across different RNA types.

---

### Official Review · Reviewer_NTRe · 2025-11-01

**Soundness:** 3
**Presentation:** 3
**Contribution:** 2
**Rating:** 4
**Confidence:** 4

**Summary:**

The paper targets RNA 3D structure prediction and systematically studies how to fuse an RNA language model (LM) into an AlphaFold-3–like architecture. Under fixed data and hyperparameters, it varies only the fusion position and mechanism, reporting notable gains on RecentPDB-RNA and claiming automated SOTA.

**Strengths:**

Data and hyperparameters are fixed; only fusion position and strategy (add/concat/cross-attention; single vs multi conditioning) are varied.

On RecentPDB-RNA, improvements in TM-score and success rate over the baseline are substantial, with stratified analyses by sequence length and identity.

**Weaknesses:**

Prior work (e.g., RhoFold+, DRfold2) already integrates LMs for RNA; contributions read as engineering optimization of where to fuse rather than new mechanisms.

Protenix is used instead of proprietary AF3, and its weaker baseline raises questions about transferability to real AF3.

12.9k training samples; tiny test splits (≈67 and ≈11) with no statistical tests or confidence intervals reported.

Performance degrades on low sequence identity (<0.5) and long sequences (>400 nt); automated methods still trail human expert solutions.

**Questions:**

Why freeze the RNA LM? What performance is forfeited by not finetuning it?

Can you disentangle LM vs MSA contributions with a full 2×2 ablation (with/without LM × with/without MSA) and report variance/CI?

Why does “single conditioning + add” at mid/late layers work best? Please provide theoretical or empirical evidence (e.g., attention/gradient flow/information bottleneck analyses).

Given the use of Protenix, what evidence shows the fusion strategy carries over to real AF3 (or other AF3 replications)?

What are the primary sources of error versus human-guided Vfold (data coverage, long-range constraints, inductive bias)? Can you provide an error taxonomy to localize the gap?

---

> ### Author Response · Authors · 2025-11-24
> **Response to Reviewer NTRe # 1**
>
> **Response to Weakness 1**:
>
> Thank you for this comment. RhoFold+ and DRfold2 both replace the Input Embedding Module in AF2/3-like architecture with an RNA LM and are trained on RNA structure data only. Instead of training a model for RNA from scratch, we aim to quickly adapt the Protenix model trained on Protein, RNA, DNA etc for better RNA 3D structure prediction by improving the RNA embeddings through the incorporation of an RNA LM. RhoFold+ was trained for 300k steps, DRfold2 was trained for 600k steps, while our model was trained for 4.4k steps only. Our contribution lies in fast adaptation of the AF3-like model for better RNA 3D structure prediction.
>
> **Response to Weakness 2**:
>
> Thank you for the comment. We could not use the proprietary AF3 due to its prohibited use policy. Instead, we used Protenix, an open-source AF3 reproduction under the Apache 2.0 license, which allows adaptation and fine-tuning while maintaining architectural and functional equivalence for reproducible research.
>
> **Response to Weakness 3**:
>
> Thank you for this insightful comment. We acknowledge that RNA 3D structure data are inherently limited. Although the training set contains 12.9k samples, it represents only 2.7k unique sequences. We agree that the test sets are small, reflecting the current availability of high-quality RNA structures. To improve rigor, we have added statistical significance tests in **Table 2** in the revised manuscript, and we are actively expanding the test set with newly released PDB structures to further strengthen the evaluation.
>
> **Response to Weakness 4**:
>
> Thank you for this insightful comment. We agree that reduced performance on low sequence identity (<0.5) and long RNA sequences (>400 nt) highlights key limitations of current RNA 3D prediction methods.
> 1. The degradation on low-identity sequences reflects the model’s limited generalization ability to out-of-distribution data,  which was also observed in DRfold2. This underscores the need for broader and more diverse training data to improve robustness.
> 2. Predicting large RNAs is inherently difficult due to their combinatorial structural complexity, long-range dependencies, sparse evolutionary information, and scarcity of experimental data (~29% in our training data). These factors make accurate modeling substantially harder than for smaller RNAs and remain a shared challenge for both automated methods and human expert-guided approaches, as shown in the following table.
>
> | Model                                     | nt ≤ 400 (7) | nt > 400 (4) | Overall (11) |
> |-------------------------------------------|-----------|-----------|----------|
> | Vfold (human expert) from CASP16 website  | 0.558     | 0.359     | 0.486    |
> | Vfold Pipeline*                            | 0.289     | /         | 0.289    |
> | RLM-aug Protenix w/ MSA (single cond., add) | 0.529     | 0.235     | 0.422    |
> **Table 1.** Comparison of TM-score on short and long RNA sequences on CASP16-RNA. Vfold Pipeline fails to predict 3D structures for 5 targets (1 short and 4 long).
>
> In practice, human experts can iteratively refine computational predictions by incorporating domain knowledge. Improved automated methods provide a stronger starting point for such expert-guided refinement; however, human-in-the-loop approaches remain difficult to scale, as they require substantial expertise and manual intervention.
>
> **Response to Question 4:**
>
> Thank you for this question. Among AF3 replications such as Protenix, Bolz-1, and Chai-1, none currently utilize MSAs for RNA, making their architectures and input settings effectively similar in the RNA case. Therefore, the training dynamics and fusion behavior observed in Protenix are expected to generalize well to these AF3-like models.
>
> **Response to Question 5:**
>
> Thank you for this insightful question. As shown in Table 1 above, the gap between Vfold (human expert) and Vfold Pipeline highlights the crucial role of human expertise in RNA structure prediction. Human experts can iteratively refine predictions using domain knowledge and experimental data, while automated models remain constrained by their architectural design and limited data coverage.
> The remaining performance gap primarily arises from:
> * Data coverage limitations: High-resolution RNA structures in the PDB are far fewer than those for proteins, restricting the progress of data-driven deep learning methods, as noted in “When will RNA get its AlphaFold moment?” (Schneider et al., 2023, NAR). Moreover, large RNAs account for only 29% of RNA3DB, further limiting modeling accuracy for long sequences.
> * MSA quality: RNA MSAs are generally shallower and less informative than protein MSAs, providing weaker evolutionary constraints.
> * Inductive bias limitations: Current models cannot model multiple conformations per sequence (with RNA3DB showing ~5 structures per sequence on average) and struggle to generalize to structurally dissimilar folds.

---

> ### Author Response · Authors · 2025-11-24
> **Response to Reviewer NTRe # 2**
>
> **Response to Question 1**:
>
> Thank you for this suggestion. We chose to freeze the RNA LM because the fine-tuning dataset is relatively small, and updating all LM parameters would likely lead to overfitting. The RNA LM (650M parameters) is even larger than the Protenix backbone (386M parameters), making full joint training impractical with the available data. Similar design choices are adopted in ESMFold (Lin et al., Science, 2023), RhoFold+, and DRfold2, where pretrained LMs are also kept frozen during structure model training.
>
> To further investigate, we performed LoRA fine-tuning (rank = 8) on the LM within the RLM-aug Protenix (single conditioning, add) architecture, introducing approximately 1.4 M additional trainable parameters. Since the LoRA adapters are newly initialized, we used a two-stage training schedule:
> * Stage 1: train LoRA adapters and the projection matrix from LM to Protenix’s representation space (lr = 1e−3, frozen Protenix backbone, max_steps=800).
> * Stage 2: jointly train adapters and Protenix (lr = 1e−4).
> All other hyperparameters were kept identical to those described in the Appendix. As shown in the following table, the LoRA-finetuned variant exhibited a notable performance drop relative to the frozen model, suggesting that partial or full LM fine-tuning does not improve performance under current data limitations, or may require extensive hyperparameter tuning to realize its potential.
>
> | Model                                      | LM Finetuning Setting | TM-score | #Success |
> |--------------------------------------------|------------------------|-----------|-----------|
> | RLM-aug Protenix w/ MSA (single cond., add)  | Frozen                | 0.438     | 14        |
> | RLM-aug Protenix w/ MSA (single cond., add)  | LoRA                  | 0.383     | 8         |
> | RLM-aug Protenix w/o MSA (single cond., add) | Frozen                | 0.389     | 12        |
> | RLM-aug Protenix w/o MSA (single cond., add) | LoRA                  | 0.368     | 5         |
> **Table 2.** Effect of LM finetuning on model performance on RecentPDB-RNA.
>
> **Response to Question 2**:
>
> Thank you for this constructive question. We performed a 2×2 ablation study examining the effects of MSA and RNA LM embeddings when fine-tuning Protenix on the same dataset and evaluating on RecentPDB-RNA (with an 80% sequence similarity cutoff).  As shown in **Figure 2** in our revised manuscript, using either MSA alone or RNA LM alone does not improve performance relative to the finetuned Protenix without MSA, whereas combining both MSA and RNA LM embeddings achieves a significant improvement, indicating strong complementarity. This is very interesting. We hypothesize that this synergy arises because both MSA and RNA LM embeddings encode evolutionary information, but in distinct feature spaces. Protenix is originally optimized to leverage MSA-based features (from protein training), so introducing RNA LM embeddings alone without MSA makes it difficult for the model to leverage these heterogeneous representations. When used together, however, the LM embeddings regularize and enrich the MSA-derived signals, leading to improved generalization. We will include detailed analyses of this interaction in the revised version.
>
> **Response to Question 3**:
>
> Thank you for this insightful question.
>
> The additive fusion approach follows the principle of feature-wise linear modulation (FiLM) (Perez et al., AAAI 2018), which selectively adjusts internal representations. It introduces the fewest newly initialized parameters compared to concatenation or cross-attention, making optimization more stable and efficient.
>
> Single conditioning works best because it minimally perturbs the MSA and Pairformer information flow while injecting complementary contextual information directly into the Diffusion Conditioning Module. Injecting RNA LM embeddings too early (at the input or initial representation stage) would require the signal to propagate through all 48 × N_recycle layers before reaching diffusion conditioning, increasing the risk of gradient vanishing. In contrast, injecting the embeddings at the diffusion conditioning stage preserves the original structural signal from the MSA and Pairformer, while allowing the LM embeddings to modulate the features selectively for structure generation. It probably raises a question of whether an LM representation should have some structure inductive bias for the diffusion model to predict the 3D structures. Interestingly, a recent study, SimpleFold (Wang et al., arXiv 2025, Apple), finds that triangle attention or pair representation biases in AlphaFold are not necessary for effective structure prediction, supporting our empirical observation that simpler, additive conditioning at later stages can suffice.
>
> Overall, the single-conditioning + additive fusion strategy provides a balanced way to integrate new contextual information without disrupting existing structural priors.

---

### Author Response · Authors · 2025-12-02

We thank the Area Chair for overseeing the review process and all the reviewers for their thoughtful and constructive feedback. We have carefully incorporated all major suggestions and substantially strengthened the manuscript. Key revisions include:
* Enforcing a stricter evaluation split (<=80% sequence identity),
* Adding statistical significance tests,
* Conducting new ablations (LM $\times$ MSA, effect of MSA quality in inference, LoRA finetuning, sampling analysis),
* Clarifying dataset construction and baseline usage, and
* Explaining more clearly why the recommended fusion strategy works

The key contribution of our work is a systematic and controlled study of how RNA language model representations can be effectively integrated into AF3-like architectures, leading to a clear, reproducible, and empirically validated recipe for improving RNA 3D structure prediction – especially on low sequence identity and long-sequence RNAs where current methods struggle.

We believe these revisions address the reviewer’s concerns and improve both the clarity and rigor of the work. We hope the strengthened manuscript will meet the expectations for acceptance.

---

### Meta-Review · Area_Chair_T6KT · 2026-01-07

**Summary:**

The reviewers are concerns about leakage, weak empirical performance, and a lack of either novelty or insight. I think that while the experiments are well-motivated, clear, and executed well, the bar for this type of paper is either very strong empirical performance OR new mechanistic or biological insight. This paper makes progress towards both, but ultimately does not clear either bar.

**Reviewer Concerns:**

**Weak overall performance compared to human-in-the-loop methods, especially on long or low-homology RNAs**

The rebuttal clarifies that the contribution is primarily about systematically demonstrating that LM fusion and MSA information can improve end-to-end predictions and that human-in-the-loop methods are not scalable. While the paper would definitely be stronger if it outperformed human-in-the-loop methods, I agree that it is still possible to find impactful insights without that. The empirical results would also be stronger if more of the performance gains came from the LM fusion instead of from vanilla finetuning on additional data.

**Possibility of train-test leakage**

The authors filter the test set by sequence homology, clarify that using the existing structure-based split is not feasible because many of the structures would be in the Protenix train set, and point to examples of other works that use sequence homology-based splits. I would have liked to see a stratification of the results by structural similarity in addition to by sequence homology.

**No insight into why this fusion method is effective or why MSAs only help when combined with the best LM fusion**

The rebuttal and revision claim that LM embeddings help the model utilize evolutionary information from MSAs. This is plausible, but at best speculative. This could have been further improved by highlighting when LM embeddings help the most (shallow MSAs, long proteins), but these conclusions are also weakened by the small sample sizes. This is not necessarily the authors' fault (there simply is not much RNA structure data), but remains a weakness of the paper.

**Only one LM tested, the LM weights are frozen, and limited analysis of the effect of the LM training data**

The rebuttal points out that freezing LM weights is the standard practice even for proteins, which have much more available structure data for finetuning, and shows that unfreezing the LM weights leads to overfitting and worse performance. Given the lack of a step change in empirical performance, an alternative route to an impactful paper could have been to systematically analyze the effects of LM architecture or data choices on performance, but this analysis was not performed. One of the rebuttals claims that a revision will include additional LMs, but I don't see it in the revision.

**Reviewer Scores:**

NTRe: 4 -> 6
JaFT: 6 -> 6
QsMV: 6 -> 6
Qgkw: 0 -> 2

---

### Decision · Program_Chairs · 2026-01-26

Reject